# Universal microbial reworking of dissolved organic matter along environmental gradients

Erika C. Freeman [1] ✉, Erik J. S. Emilson [2,3], Thorsten Dittmar [4,5], Lucas P. P. Braga [1], Caroline E. Emilson[2], Tobias Goldhammer [6], Christine Martineau[7], Gabriel Singer[8] & Andrew J. Tanentzap [1,3]

Soils are losing increasing amounts of carbon annually to freshwaters as dissolved organic matter (DOM), which, if degraded, can offset their carbon sink capacity. However, the processes underlying DOM degradation across environments are poorly understood. Here we show DOM changes similarly along soil-aquatic gradients irrespective of environmental differences. Using ultrahigh-resolution mass spectrometry, we track DOM along soil depths and hillslope positions in forest catchments and relate its composition to soil microbiomes and physico-chemical conditions. Along depths and hillslopes, we find carbohydrate-like and unsaturated hydrocarbon-like compounds increase in abundance-weighted mass, and the expression of genes essential for degrading plant-derived carbohydrates explains >50% of the variation in abundance of these compounds. These results suggest that microbes transform plant-derived compounds, leaving DOM to become increasingly dominated by the same (i.e., universal), difficult-to-degrade compounds as degradation proceeds. By synthesising data from the land-to-ocean continuum, we suggest these processes generalise across ecosystems and spatiotemporal scales. Such general degradation patterns can help predict DOM composition and reactivity along environmental gradients to inform management of soil-to-stream carbon losses.

The fate of carbon exported from soils into aquatic ecosystems is a poorly understood component of the global carbon cycle. Soils store at least twice as much carbon as the atmosphere[1] and are expected to absorb more than one-third of anthropogenic emissions[2]. However, >15% of the net carbon added to soils annually from decomposing plant litter and roots is leached into aquatic systems as dissolved organic matter (DOM)[3]. Once in water, much of the DOM pool is highly reactive[4], potentially returning large quantities of carbon to the atmosphere as $CO_2$ or $CH_4$ and offsetting terrestrial carbon sequestration[5].

The distance from its terrestrial sources can predict how much DOM degrades in aquatic ecosystems[6]. Terrestrial DOM is initially

[1]Ecosystems and Global Change Group, Department of Plant Sciences, University of Cambridge, Cambridge CB2 3EA, UK. [2]Natural Resources Canada, Canadian Forest Service, Great Lakes Forestry Centre, 1219 Queen St. E., Sault Ste, Marie, ON P6A 2E5, Canada. [3]Ecosystems and Global Change Group, School of the Environment, Trent University, Peterborough, ON K9L 0G2, Canada. [4]Institute for Chemistry and Biology of the Marine Environment, University of Oldenburg, 26129 Oldenburg, Germany. [5]Helmholtz Institute for Functional Marine Biodiversity, University of Oldenburg, 26129 Oldenburg, Germany. [6]Department of Ecohydrology and Biogeochemistry, Leibniz-Institute of Freshwater Ecology and Inland Fisheries, Mueggelseedamm, 301 Berlin, Germany. [7]Natural Resources Canada, Laurentian Forestry Centre, 1055 Du P.E.P.S. Street, P.O. Box 10380, Québec G1V 4C7, Canada. [8]Department of Ecology, University of Innsbruck, Technikerstrasse 25, 6020 Innsbruck, Austria. ✉e-mail: erika.freem@gmail.com

dominated by a few spatially heterogeneous biomolecules, such as lignin-derived polyphenols that reflect local plant species composition[7]. Continuous transformation and remineralisation of this DOM along merging flowpaths produce an increasingly homogeneous pool of compounds downstream[8–10]. Compounds with structural features such as carboxylic-rich alicyclic moieties, material derived from linear terpenoids, and carotenoid degradation products dominate this increasingly homogeneous pool[11]. As these compounds occur everywhere, that is, in all samples, they are termed "universal"[12] or "core"[13]. The convergence towards a DOM pool dominated by universal compounds is known as a "degradation cascade". Later stages along a degradation cascade should also have a greater proportion of shared compounds[12], such as measured with molecular β-diversity, but this idea remains untested. All else being equal, the amount of time DOM is exposed to microbial and photochemical processing[14] is likely a unifying explanation for the degradation cascade[15]. Residence time correlates with carbon decay rates in marine sediments[16,17], bioassays[18], and inland waters[4,10]. As many universal compounds are degraded slowly by microbes[19], DOM pools that are homogenised later along a degradation cascade can ultimately provide a persistent carbon store.

Universal DOM pools can result from common synthetic pathways or a chain of similar degradation steps[6,12,20–22], but how the importance of these mechanisms changes along diverse environmental gradients remains unknown[15]. Unlike along depth profiles in oceans, processing of DOM during vertical soil passage does not consistently converge to low-molecular-weight, recalcitrant compounds[23]. Degradation of DOM is instead characterised by increasing molecular weight. The process generating this increase in weight likely reflects the extracellular microbial decomposition of large macromolecular plant material into smaller molecules, for example, the production of simple unsaturated oligogalacturonates after pectin degradation[24]. These small metabolites are then transformed to larger microbially-derived compounds, namely complex polysaccharides associated with microbial tissues and products, such as chitin or glucans[23,25]. Many microbes synthesise carbohydrates, hydrocarbons, and lipids in this way[26,27]. This process partly reflects the function of semipermeable cell membranes, where diffusion is restricted to molecules of low molecular weight, but these low-molecular-weight substances are subsequently elongated and incorporated into larger cell structures[26]. Degradation also varies with compound concentration[28], abiotic processes like sorption and desorption to minerals[29], hydrological pathways[30], and microbial trait diversity and energy supply[31]. Thus, DOM composition, microbial metabolism, and environmental and ecosystem properties interact to stabilise carbon[32]. Understanding how these processes influence DOM degradation along different flowpaths is necessary to ensure land-based carbon sequestration efforts are not offset downstream.

Here we asked if the molecular composition of DOM changes similarly through soil depth and along hillslopes with different environmental conditions and thus potential degradation processes. We worked across four replicate headwater catchments in northwestern Ontario, Canada (Fig. S1). We advanced previous studies by focusing on soil-water flow upstream of the headwater-ocean continuum, i.e., a natural but neglected extension of the riverine continuum[33]. We tracked DOM from 5 to 60 cm soil depth at each of shoulder, back, toe, and foot hillslope positions and into streams using Fourier-transform ion cyclotron resonance mass spectrometry (FT-ICR-MS) (Fig. 1a). We paired FT-ICR-MS with shotgun metatranscriptomic sequencing and metabolic measurements to reconstruct the function of microbial communities. Our fully factorial depth-by-hillslope design allowed us to test how DOM originating from the same source material (i.e., within each position) changed along contrasting environmental gradients and into headwater streams.

Although we expected DOM to become increasingly homogeneous across both soil depth and hillslope, consistent with a degradation cascade[12], we tested two contrasting hypotheses for how

homogenisation occurred and the underlying mechanisms. The first hypothesis was that the extent of homogenisation and underlying physico-chemical conditions or microbial processes would differ. With depth, we predicted the DOM pool should reflect preferential consumption and transformation by microbial processes[23]. We therefore expected DOM sources (i.e., plant litter) to be relatively consistent through depth with less hydrological mixing of sources than between hillslope positions. Along the hillslope, we predicted DOM should reflect different sources because of hydrological mixing from different positions that vary in moisture, erosion, vegetation type and rooting depth. Therefore, we expected microbial processing along the hillslope should be relatively less important than with increasing soil depth because of the relatively large variation in DOM sources. In contrast, our second hypothesis was that DOM composition could be homogenised similarly between the two spatial gradients of depth and hillslope. This hypothesis predicts that universal processes, such as the duration of microbial processing, shape similar DOM composition despite environmental differences along the degradation cascades. Our results now implicate common microbial metabolic processes in transforming non-universal compounds, leaving behind a universal pool of DOM along the soil-headwater continuum, and suggest that this process generalises across environmental and spatiotemporal gradients.

## Results and discussion
### Degradation of DOM across soil depth and hillslope
Consistent with the predicted increase in universal compounds from headwater to ocean[12], we found upland soils had fewer universal compounds than aquatic samples. We detected 12,487 peaks and assigned 9327 unique molecular formulae (75% of peaks detected), more than twice that observed in a previous soil study using the identical FT-ICR-MS method and instrument[23], suggesting we representatively sampled the DOM pool. We attributed this result to high extraction efficiencies for DOM (mean 69% ± 6% s.d.; see the "Methods" section), technical improvements in the detection cell, and advancements in molecular formula assignment[34]. Despite the many detected formulae, only 13% occurred in all samples, that is, were universal, compared with between 47 and 87% in a synthesis from headwaters to oceans using the same method and instrument[12]. Of the universal compounds, 79% were classified[35] as lignin-like based on the similarity of their elemental compositions to known biomolecules (Fig. 1b). Tannin-like (representing phenol derivatives) and condensed hydrocarbon-like compounds were the next most abundant classes based on counts of molecular formulae, accounting for 9%, and 6% of all formulae, respectively. Our results were not due to missing high-molecular-weight substances (HMWS) commonly detected in soils[36] but outside the analytical window of FT-ICR-MS[37]. Using size-exclusion chromatography to quantify DOM fractions, HMWS were detected in only 13% of soil pore water samples and, when found, contributed, on average, only 8% (95% confidence interval, CI: 3–12%) to the total dissolved organic carbon concentration (Table S1).

Throughout the soil depth profile, the DOM pool converged upon a universal pool but not along landscape hillslope positions. Consistent with our predictions, the percentage of universal compounds increased from an estimated mean of 20.9% (95% CI: 19.0–22.9%) of all formulae at 5 cm to 23.9% (21.9–26.1%) at 60 cm (Fig. 1c, generalised linear model: $z = 2.1$, $p = 0.036$, df = 57). There was no change in the proportion of universal compounds between the shoulder position and streams (Fig. 1d; Table S2), as expected if hydrological mixing was important along the hillslope gradient. However, when we measured the proportion of signal intensity (i.e., relative abundance) attributed to universal compounds, the DOM pool was similarly homogenised along both soil depth and hillslope gradients. The relative abundance of universal compounds increased by 9.4% (95% CI: 5.9–12.9%, linear model: $t = 5.5$, $p < 0.001$, df = 57) from an estimated mean of 54.3%

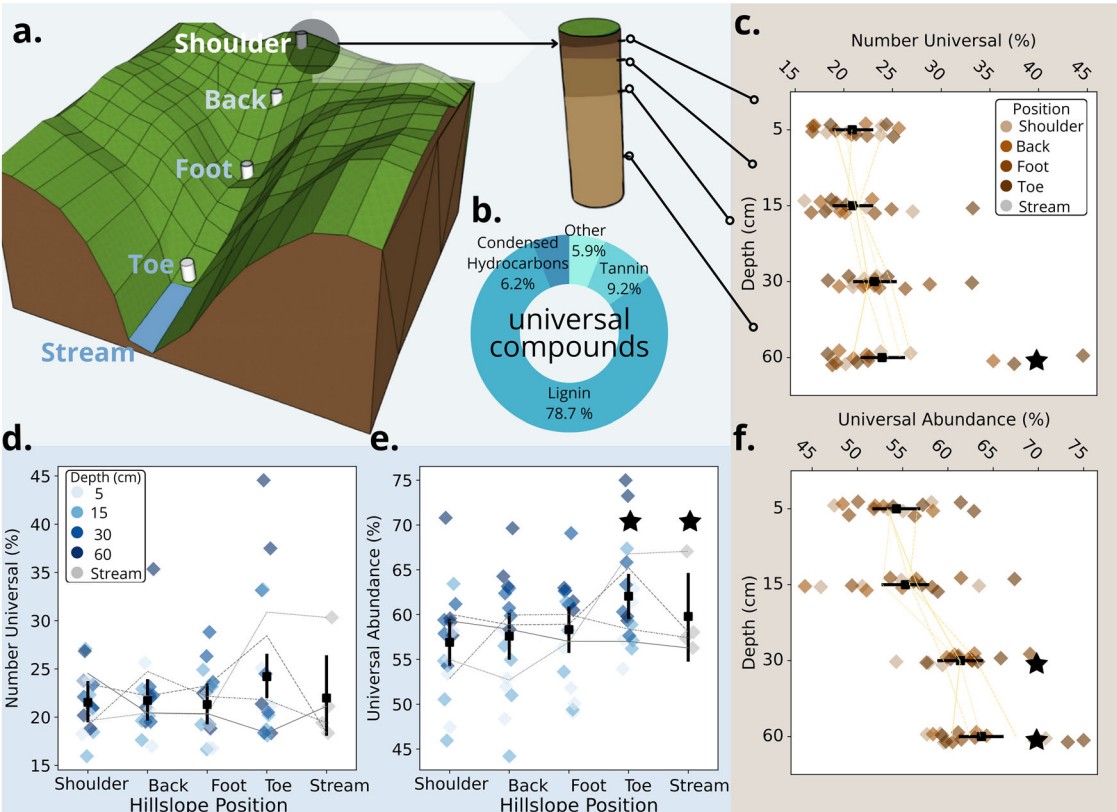

**Fig. 1 | Dissolved organic matter converges to a universal pool across soil depth and hillslope. a** Sampling design showing the location of soil lysimeters at shoulder, back, foot, toe hillslope positions and the headwater stream in a study catchment. At each hillslope position, we sampled soil at 5, 15, 30, and 60 cm depth. **b** Relative contribution of classes to the universal compound pool (*n* = 1216 molecular formulae). Lignins and tannins refer to molecular formulae that are phenol derivatives. **c–f** Composition of universal compounds, which are defined as those present in all samples. Points (diamonds) are percentage of compounds and relative abundance (i.e., sum of normalised signal intensities) comprised by universal compounds along the soil depth (**c** and **f**, respectively) and hillslope (**d** and **e**, respectively) gradients. Gradients of colour are increasing soil depth or hillslope position. Estimated marginal means ± 95% CI denoted by black squares were averaged across catchments and either depth or hillslope positions. Grey and brown dotted lines are mean values for catchment replicates for each position and depth, respectively (*n* = 4). Black stars denote hillslope positions and depths that are statistically different from either the shoulder or 5 cm samples, respectively, based on Tukey-adjusted *p* values estimated from a generalised linear model (Table S2).

(51.8–56.8%) at 5 cm to 64% (61.4–66.0%) at 60 cm (Fig. 1e). There was a similar 8.1% (6.4–9.7%, *t* = 3.0, *p* = 0.029, df = 57) relative increase in universal compounds from 56.9% (54.4–59.3%) at the shoulder to 59.8% (54.9–64.5%) at the stream, respectively (Fig. 1f). Universal compounds identified as those occurring in all our samples had similar molecular properties to literature definitions of degradation end-products that were independent of our sample set (Fig. S2). We found similar results when we matched our molecular formulae to those considered universal[22] across aquatic ecosystems, again, likely because they reflect end-products of degradation (Table S2).

Convergence towards a universal DOM pool was due to the preferential degradation of subsets of compounds actively reworked by microbes, leaving behind the universal pool of compounds that was observed across all samples. To understand better the biogeochemical processes driving the shift towards universal DOM, we deconstructed universal compounds into putative compound classes along both soil profile and hillslope gradients (Table S3). Universal lignin-like compounds (representing phenol derivatives) increased from an estimated mean of 40.1% (95% CI: 37.4–42.8%) of the relative abundance of all molecular formulae to 52.9% (50.2–55.6%, linear model: *t* = 7.8, *p* < 0.001, df = 57) from 5 to 60 cm depth. These same compounds increased from an estimated mean of 43.6% (40.9–46.2%) to 48.4% (43.1–53.8%, *t* = 4.8, *p* < 0.001, df = 57) from the shoulder position into the stream (Table S3, Fig. S3). The declines in relative abundance of universal molecular formulae from other compound classes were not

large enough to account entirely for the increased representation of universal lignin-like compounds (Table S3, Fig. S3). Other non-universal compounds must have also become proportionally less abundant, such as if they were preferentially removed, for plant-derived lignin compounds to become increasingly represented in the DOM pool.

To explain further the shift towards universal compounds across the two gradients, we examined the relative abundance and intensity-weighted mass of non-universal compound classes. The relative abundance of non-universal tannin-like and condensed hydrocarbon-like compounds together declined by an estimated mean of 12.9% (95% CI: 8.9–16.9%, linear model: *t* = 5.5, *p* < 0.001, df = 57) from 23.6% (16.9–20.8%) to 10.7% (7.9–13.5%) and by 13.3% (9.93–16.6%, *t* = 4.0, *p* = 0.002, df = 57) from 21.2% (18.4–15.2%) to 14.7% (9.1–20.2%) along the depth and hillslope gradients, respectively (Table S4, Fig. S4). Both these compound classes tend to reflect plant material rather than microbial products[38,39]. Although other non-universal classes showed small average increases (<4%; Table S4, Fig. S4), these could not balance the declines in tannin-like and condensed hydrocarbon-like compounds, as expected if plant material was generally being degraded. In support of the degradation of specific compound classes causing a shift towards universal DOM, we found that intensity-weighted mass of non-universal compounds increased along the depth and hillslope gradients (Fig. 2). These changes were similar between gradients despite

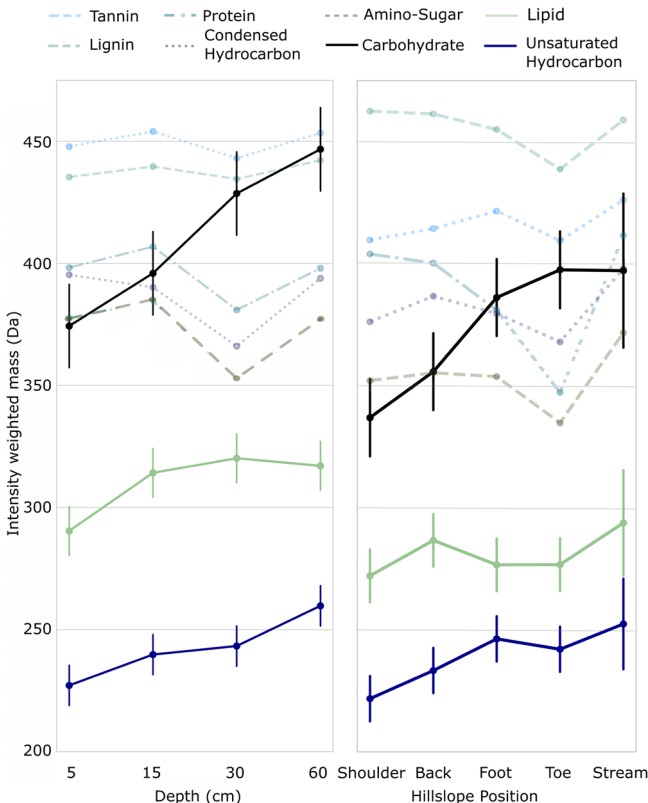

**Fig. 2 | Compound classes shift in mass across soil depth and hillslope position.** Mean estimated molecular mass (±95% CI) of dissolved organic matter for each compound class when universal compounds were removed from calculations, that is, we plot only non-universal compounds. Molecular formulae were grouped into compound classes based on their atomic ratios. Solid lines are those with statistically significant differences between 5 and 60 cm depths or shoulder and stream positions based on Tukey-adjusted *p* values estimated from a generalised linear model (*n* = 16 samples, Table S5). Errors for non-statistically-significant compound classes are presented in Table S5.

differences in environmental conditions. From 5 to 60 cm depths, carbohydrate-, unsaturated hydrocarbon-, and lipid-like compounds increased in weighted mass by an estimated mean of 19.5% (95% CI: 10.8–21.6%, $t = 6.0$, $p < 0.001$, df = 57) from 374 Da (357–391 Da) to 447 Da (430–464 Da), 14.2% (7.8–17.7%, $t = 5.2$, $p < 0.001$, df = 57) from 253 Da (242–263 Da) to 289 (279–299 Da), and 10.6% (4.7–15.0%, $t = 3.9$, $p < 0.001$, df = 57) from 290 Da (278–301 Da) to 321 Da (310–333 Da), respectively. Carbohydrate- and unsaturated hydrocarbon-like compounds also increased consistently by 11.9% (6.0–23.3%, $t = 3.4$, $p = 0.001$, df = 57) from 377 Da (360–394 Da) to 442 Da (408–476 Da) and 8.7% (3.9–20.2%, $t = 3.0$, $p = 0.005$, df = 57) from 252 Da (242–262 Da) to 274 (264–284 Da), respectively, across hillslope positions from shoulder into the stream (Table S5). Although lipid-like compounds also increased from shoulder into the stream, they did not do so across the other hillslope positions like the carbohydrate- and unsaturated hydrocarbon-like classes (Table S5), potentially reflecting the larger sizes of lipids produced by aquatic primary producers[40]. The strength of the intensity-weighted mass shifts of non-universal compounds were large enough to increase intensity-weighted mass of all compounds along both soil profile and hillslope gradients (Table S6, Supplementary Results). Together, these results suggested that either larger mass compounds were being conserved or that new, heavier compounds were being created from lighter precursors. The latter scenario was expected if microbes were processing plant-derived compounds by degrading them into

smaller compounds that were subsequently incorporated into larger microbial products[23].

Environmental controls over the compound classes were identified by comparing soil depth and hillslope. As expected[41,42], absolute organic carbon concentrations declined from 5 to 60 cm and from shoulder to stream by an estimated mean of 360% (95% CI: 274–474%, linear model: $t = 9.4$, $p < 0.001$, df = 57) and 71% (66–87%, $t = 6.5$, $p < 0.001$, df = 57), respectively (Fig. 3a). The decline in DOC was associated with greater microbial productivity along the hillslope (Fig. 3b; Table S7, $t = 7.2$, $p < 0.001$, df = 25), suggesting that it could reflect microbial carbon consumption and not simply lower inputs of carbon at depth. Consistent with this result, we found a shift from humic-like to low-molecular-weight, microbial-derived carbon along the hillslope using the larger analytical window of size-exclusion chromatography (Fig. 3c, Table S7). The ratio of humic-like substances to low-molecular-weight carbon chromatographic fractions decreased by an estimated mean of 57.7% (17.0–78.4%, $t = 2.6$, $p = 0.013$, df = 57) towards the streams (Fig. 3c). By contrast, dissolved organic carbon to total nitrogen (C:N) concentrations in soil pore water decreased with depth, also as expected because of microbial processing[43], by an estimated mean of 119% (53–213%, $t = 4.4$, $p < 0.001$, $n = 57$). There was no change in C:N ratios with hillslope position (Fig. 3d). Alongside the evidence of microbial consumption of dissolved organic carbon (Fig. 3a), nitrogen-rich proteins identified by FT-ICR-MS strongly declined from shoulder to toeslope (Fig. 2b). Nitrogen in the form of proteins was likely selectively adsorbed by clays[44] that accumulate at the bottom of hillslopes[45]. Thus, these results suggest that the environmental conditions differed between the depth and hillslope gradients (Fig. 3), yet the DOM pool consistently converged upon a universal compound pool (Fig. 2). Microbial processing may explain homogenisation of DOM along the two degradation cascades, although the exact metabolic pathways may have differed.

## Microbial processing explains shifts towards a universal DOM pool

To understand why non-universal compounds in the different classes reflected increased microbial reworking, we partitioned the variation in their relative abundance using redundancy analysis. We compared the importance of spatial variation along both soil depth and hillslope position, including autocorrelation among sample locations, with environmental variables as sources of variation. We measured 475 environmental variables related to the gene activity of carbohydrate-degrading enzymes, the activity of extracellular enzymes or level of carbohydrate substrate utilisation, and soil physico-chemical conditions (Table S8). To prevent overfitting, we reduced the environmental variables by removing highly inter-correlated parameters (|$r$| > 0.80) and then further selected predictor variables using an automated permutation-based procedure (see Methods). For carbohydrate- and unsaturated hydrocarbon-like compounds that showed evidence of progressive reworking, that is, increased average weighted molecular mass (Fig. 2), we found that the spatial gradients (depth and position) were more important than for other compound classes (Fig. 4a). Overall, we likely identified many of the most important variables structuring DOM composition as our analysis explained 66–91% of the variation in the composition of compound classes (Fig. 4b).

Most of the environmental variation in DOM composition was due to the potential and realised activity of microbial communities, particularly for compound classes that reflected increased processing along the soil depth and hillslope gradients. We explained differences in the processing of DOM by quantifying the contribution of each environmental variable to the variance partition analysis. This analysis identified 62 environmental variables that were important for explaining DOM composition across the 8 compound classes (Table S8). For non-universal carbohydrate- and unsaturated hydrocarbon-like compounds, which showed increased reworking

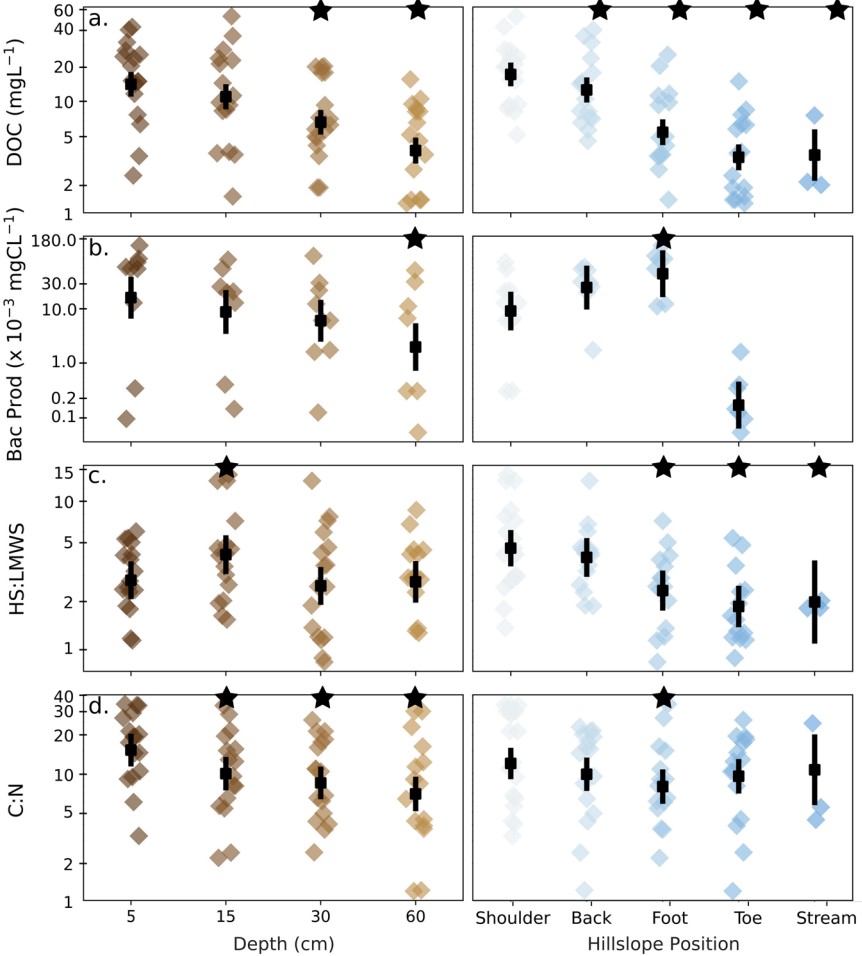

**Fig. 3 | Carbon concentration and quality change along soil depth and hillslope.** We measured soil pore water for **a** dissolved organic carbon (DOC) concentrations ($n = 16$ samples). **b** bacterial protein production (Bac Prod; no stream sample was taken); **c** fractions of humic substances (HS) to low-molecular-weight substances (LMWS); and **d** the ratio of total organic carbon to total nitrogen (C:N) concentrations. Points are estimated means ± 95% CIs. Black stars denote hillslope positions and depths that are statistically different from either the shoulder or 5 cm samples, respectively, based on Tukey-adjusted $p$ values estimated from a generalised linear model (Table S7).

with both depth and hillslope, variables associated with realised microbial activity together explained more variation in composition than for any other compound class (25 and 27%, respectively; Fig. 4b). As the relative intensity of non-universal compounds is the inverse of universal compounds, we did not repeat the variance partition analysis with the universal pool.

Carbohydrate-active enzymes (CAZymes) primarily used to break down plant-derived carbohydrates were common when we identified the environmental variables shared only by compound classes that changed along the spatial gradients. For carbohydrate- and unsaturated hydrocarbon-like compounds, the activities of four CAZymes were exclusively shared in the lists of the most important environment variables (Fig. 4b). These CAZymes encoded lignocellulolytic enzymes involved in plant cell wall degradation (auxiliary activity 1[46]), glycoside hydrolases involved in degradation of both xylan (glycoside hydrolase 43) and microbial cell walls (glycoside hydrolase 23)[47], and an enzyme involved in the breakdown of polysaccharide carbon-oxygen bonds (polysaccharide lyase 33)[48]. We also found evidence that these enzymes were expressed at a community-level as measured from enzyme activity assays. Catabolic use of two carbohydrate substrates (DL-α-glycerol phosphate and β-methyl-D-glucoside) by the microbial communities was also identified exclusively in the lists of the most important variables for carbohydrate- and unsaturated hydrocarbon-like compounds (Fig. 4b). For lipid-like compounds that also increased

consistently in molecular mass with soil depth, only a family of cellulose-binding enzymes (carbohydrate-binding module 2[49]) was shared with carbohydrate- and unsaturated hydrocarbon-like compounds (Fig. 4b). These results provide more direct evidence than previously[23] that microbial processing underlies the progression towards a universal DOM pool along spatial gradients[50].

We further found that the soil microbiome shifted in activity from processing plant- to microbial-derived OM across the spatial gradients, providing direct evidence of the microbiome's importance in generating a universal DOM pool. We tested if the transcription of genes annotated as CAZymes was differentially expressed. For CAZymes that were associated with the non-universal compound classes that increased in molecular mass along across the depth and hillslope gradients (i.e., bolded classes in Fig. 4b), we found that 9 statistically differed from 5 to 60 cm depth (Table S9). These genes included carbohydrate-binding module (CBM) 2 and glycoside hydrolase (GH) 23, 129, and 135, the latter of which was among the most important for the composition of carbohydrate- and unsaturated hydrocarbon-like compounds (Fig. 4b). All these CAZymes can degrade polysaccharides from fungal and bacterial biofilms and cell walls[51–53] and together explained 11–28% of the variation within compound classes that showed signs of processing with depth (i.e., bolded classes in Fig. 4b). As expected if OM inputs shifted from plant- to microbial-derived with increasing soil depth[23], expression of these genes increased from 5 to

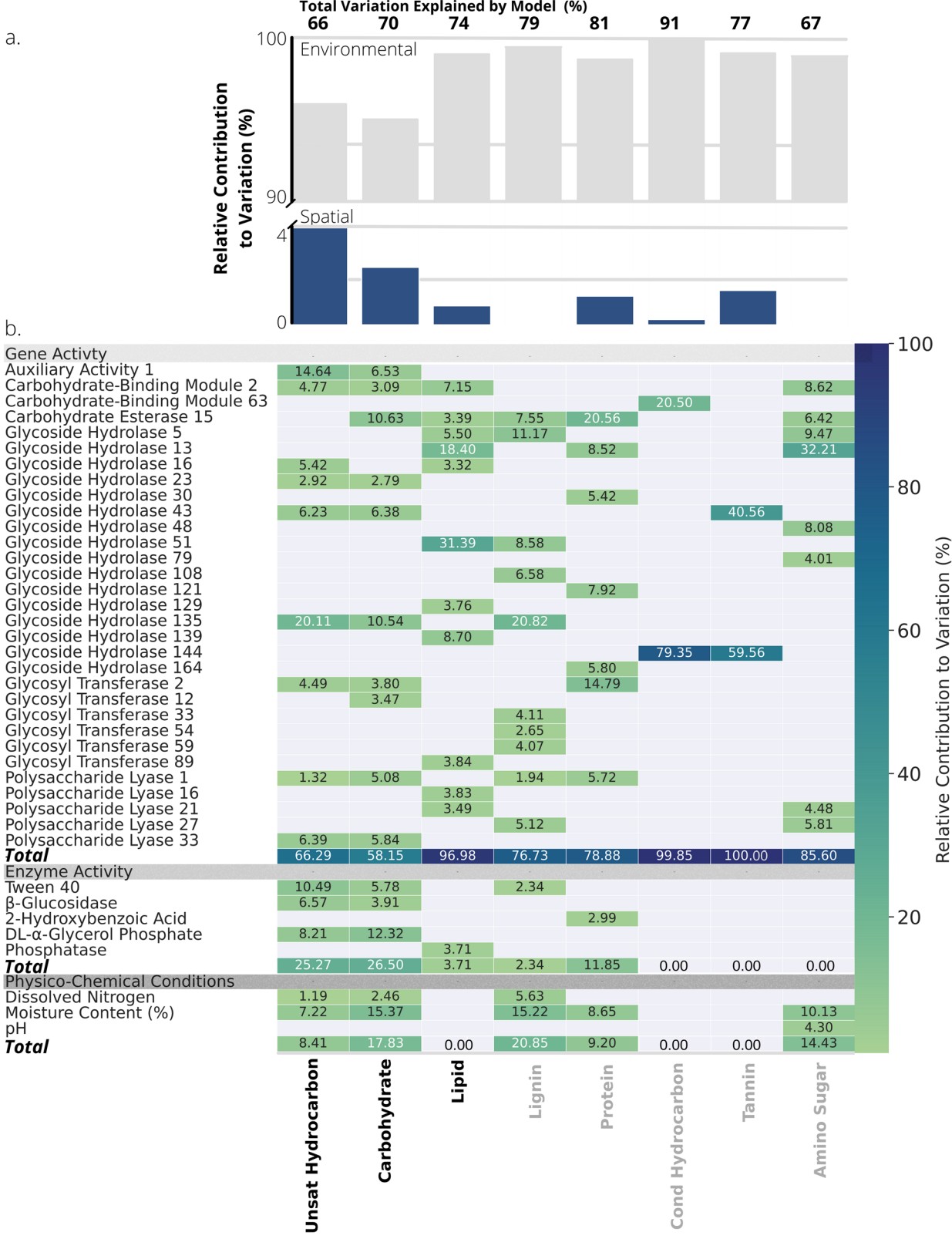

**Fig. 4 | Spatial gradients and environmental variables explain variation in the molecular composition of different compound classes. a** The relative variation explained by spatial and environmental variables in a variance partition analysis of the relative abundances of non-universal-molecular formula in each compound class. Above each bar is the absolute variation (%) explained by the model.
**b** Relative contribution of environmental predictors in explaining total variance determined by hierarchical partitioning of analysis in (**a**). Variables were grouped

according to their association with the gene activity of carbohydrate-degrading enzymes, the activity of extracellular enzymes or level of carbohydrate substrate utilisation, and physico-chemical conditions. Variation explained by a gene family was displayed if >2.5% for at least one compound class and was summed across all identified subfamilies within a family. Bolded compound classes increased in molecular mass with depth and/or hillslope (Fig. 2).

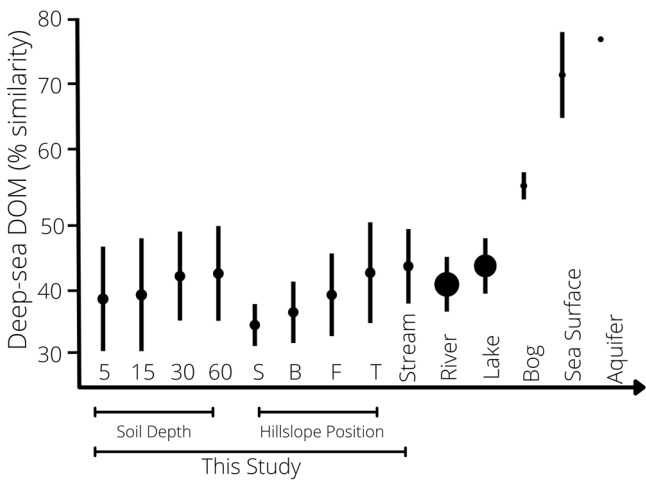

**Fig. 5 | Soil pore water is an extension of the aquatic continuum.** The mean percentage of molecular formulae shared with a deep-sea reference sample was calculated for 433 samples from 5 published FT-ICR-MS studies across a headwater-ocean continuum (Table S10). To account for differences in per-sample molecule number arising from variation in instrument resolution and sample preparation, we used a rarefaction approach employed in microbial diversity studies[58]. We randomly sampled each study 1000 times to a set of 6000 compounds. This threshold was determined based on a minimum number of molecules to adequately sample the pool of observed compounds (Supplementary Methods). Points are means ± standard deviations of observations. Sizes of points scale with the number of observations at each position along the land-ocean continuum ($n = 2–116$, aquifer to river). Results for soil depth and hillslope position were averaged across sites for each respective gradient. S shoulder, B backslope, F footslope, T toeslope.

60 cm by a mean (95% CI) of 1.7–2.9-times (Fig. S5). By contrast, 4 CAZymes differed statistically from shoulder to toe positions (Table S9), and explained about 11%, on average, of the variation in compound classes that showed signs of processing with hillslope (Fig. 4b). Three of these CAZymes help degrade plant-derived OM[48,54,55]: GH43 subfamily 23, GH139, and polysaccharide lyase 16, and they all decreased in expression by a mean of between 3.7 and 22.6-times from shoulder to toe positions (Fig. S5). Because genes also interact, expression patterns can shift across the spatial gradients without necessarily showing monotonic increases. Therefore, we also used principal component analysis (PCA) to visualise differences in gene expression along the spatial gradients. Both depth and hillslope explained variation in gene expression as they were correlated with the first two axes of the PCA ($R^2 = 0.11$, $p = 0.029$ and $R^2 = 0.15$, $p = 0.047$, respectively). At later stages of the degradation cascades, expression shifted towards CAZymes that degrade microbial-derived OM, e.g., GH23, GH135, and CBM2 (Fig. S6). Together, these results further implicate microbes in transforming plant-derived compounds from shallower depths or higher hillslope positions into larger molecular-weight compounds microbial origin at later positions along degradation cascades.

### Generalising the persistence of DOM across the land-ocean continuum

Our study provides evidence that DOM converges towards a universal compound pool dominated by lignin- and tannin-like compounds that are left behind as microbial reworking removes components typical of shallower soil depth and higher hillslope positions. To determine if this process could apply more generally, we contextualised our observed patterns within the headwater-ocean continuum by synthesising published FT-ICR-MS data. Rather than calculate universal compounds in a global pool of samples, we measured convergence as similarity to a deep-sea reference sample to facilitate inter-study comparison.

Although the percentage of compounds shared among all samples in a study reflects convergence towards a universal pool, it is sensitive to differences among studies in molecule number, formula attributions, and sampling intensity. The deep-sea sample is the endpoint of degradation along the land-ocean continuum, so should accumulate the highest proportion of universal compounds[12]. We subsequently found samples expected to be exposed to microbial processing for longer, that is, further along the land-ocean continuum, such as deep soils and toeslope and stream positions, were most like deep-sea DOM (Fig. 5). The degree of convergence towards a universal pool was consistent with our observations for upland soils, generally, and the position of our different study depths and hillslopes (Fig. 5). These results suggest that a similar process of cumulative exposure to microbial processing may explain changes in DOM along similar spatial and temporal gradients, though the extent to which these trends are linear may vary with time[15], hydrological mixing[30], and rooting depth[23]. This process would provide the mechanistic basis for popular heuristic models like the river continuum concept[66] and soil chromatograph[56]. More sophisticated ecosystem models could now leverage this pattern in DOM composition to predict variation in DOM degradation[32] and identify potential carbon sinks[57].

By demonstrating that microbial processing changes the abundance of individual compounds along different environmental gradients, our work advances our understanding of how organic compounds accumulate and influence soil carbon sequestration. Two dominant processes likely contributed towards this result. First, universal compounds, especially those that were lignin-like, may have been selectively retained because of the energetic costs required for their degradation[59]. Second, non-universal compounds (for example, in the carbohydrate-like class) were consistently reworked by microbes. Genes encoding enzymes involved in the breakdown of plant cell walls were especially related to mass shifts of microbially produced compounds, e.g., carbohydrate-like rather than lignin- or tannin-like (Fig. 4). Our study was not designed to test a third hypothesis commonly invoked to explain DOM accumulation in oceans: compounds persist because their concentrations are too low to interact with microbes[28]. Despite homogenisation of chemical traits, DOM has been found to become more structurally diverse along the degradation cascade[12]. Individual structures may occur at concentrations that limit microbial activity[11], and our paired FT-ICR-MS, shotgun metatranscriptomic sequencing and metabolic methods approach could be applied in the future to test this idea explicitly. More generally, the reasons for the spatial variation in the degradation state of DOM that we identified here can improve soil-to-stream carbon management.

## Methods

### Field site and water sampling

Samples were collected from four forested catchments in northwestern Ontario, Canada (46° 58′ N, 84° 22′ W, altitude 375 m; Fig. S1). Within each catchment, hillslopes were partitioned into four positions: shoulder, backslope, footslope and toeslope. Classifications were based on the morphology of surface digital elevation models[60] and a Height Above the Nearest Drainage terrain model[61] that provides a spatial representation of soil-water environments[62]. At all hillslope positions, soil water was sampled in October 2019 after two months of continuous calibration sampling. Samples were collected at 15, 30 and 60 cm depths with tension lysimeters that consisted of 60-mm-long round bottom necked porous cups with an outer diameter of 48 mm and effective pore size of 1.3 μm (model 0653X01-B02M2, Soilmoisture Equipment Corp, USA). The sampling bottles were evacuated to a negative pressure of 50 kPa with a hand pump so suction pressure was ca. 50 mbar above the actual soil-water tension. At 5 cm depths, lysimeters could not be securely installed. Therefore, we sampled pore water with micro-tensiometers designed to extract fluids non-

destructively using a vacuum[63] through a 0.6 μm ceramic cup (Rhizon CSS samplers, Rhizosphere Research Products, The Netherlands). All the water samplers were installed in triplicate and pooled at each of the depth-by-position combinations to retrieve sufficient volume. Surface water from streams was grab-sampled at the bottom of each hillslope at the channel head. The hillslope design of 16 soil samples and 1 stream sample was replicated across the four catchments for a total of 68 samples.

All water was filtered within 8 h through 0.45 μm glass fibre filters (Whatman GF/F, pre-combusted 400 °C, 4 h) and treated differently depending on the eventual analyses. For fluorescence spectroscopy, samples were aliquoted with no headspace into 20 mL borosilicate scintillation vials and stored at ca. 4 °C in the dark. For size-exclusion chromatography, samples were frozen in acid-rinsed and pre-combusted 22 mL borosilicate vials with a PTFE/silicone septa and a polypropylene cap. For DOC concentrations and FT-ICR-MS, samples were acidified with 37% trace metal grade HCl to pH 2 and stored in pre-combusted 40 mL amber borosilicate vials with a PTFE/silicone septa and polypropylene cap at ca. 4 °C in the dark.

## DOM characterisation and concentration

To characterise DOM, an aliquot of acidified sample (0.1–30 mL, depending on the DOC concentration) was desalted and concentrated via solid-phase extraction (SPE) using a styrene-divinylbenzene copolymer sorbent (100 mg cartridges, Agilent Technologies, USA) using established methods[61]. The sorbent was pre-soaked in HPLC-grade methanol the night prior to extraction. Cartridges were sequentially rinsed with ultrapure water, methanol and ultrapure water acidified with HCl to pH 2. Acidified ultrapure water was stored in the same type of bottles as the procedural blanks. After loading the SPE cartridges with sample, the cartridges were rinsed with acidified ultrapure water and dried. The DOM extracts were eluted with methanol. On each day of extraction, a process blank extract was produced. Redundancy analysis confirmed that the extraction efficiency had no influence on the molecular composition of DOM (explained variation = 0.7%, pseudo-F = 0.6, P = 0.722).

We analysed the DOM extracts on a solariX FT-ICR-MS with a 15 Tesla magnet (Bruker Daltonics, Germany). The system was equipped with an electrospray ionisation source (ESI, Bruker Apollo II) applied in negative ionisation mode. We diluted methanol extracts to a final concentration of 2.5 ppm in a 1:1 (v:v) methanol:water solution before injecting 100 μL into the FT-ICR-MS instrument. For each measurement, we collected 200 scans in duplicate. An in-house mass reference list was used for internal calibration using Data Analysis Software Version 4.0 SP4 (Bruker Daltonics, Germany). The mass error of the calibration was <0.06 ppm for all samples. All measurements were done within 7 days and in random order. In-house reference samples[62] were used to confirm instrumental stability.

Masses ranging from 150 to 1000 m/z were exported from the Bruker Data Analysis software, and we assigned molecular formulas using the online formula assignment and analysis tool ICBM-OCEAN[64] (freely available at https://rhea.icbm.uni-oldenburg.de/geomol/). A method detection limit of 2 was applied to all exported masses[65]. All molecules were singly charged ions and we limited formula attributions to $C_{0-100}$, $H_{2-200}$, $O_{0-70}$, $N_{0-3}$, $S_{0-2}$, $P_{0-2}$ with a tolerance of 0.2 ppm. Only signals detected in both duplicate measurements were retained. We normalised peak intensities of the peaks with an assigned molecular formula to the sum of peak intensities. Intensity-weighted mass-to-charge ratios ($m/z_{wm}$) for each DOM sample were calculated as the sum of the product of the $m/z_i$ for each compound $i$ and their relative intensity $I_i$ divided by the sum of all intensities $m/z_{wm} = \Sigma(m/z_i I_i)/\Sigma I_i$. We also assigned molecular categories to all formula using established criteria[35]. We further defined the structural features[64] in each category to ease semantic confusion among classification schemes (Fig. S7).

A liquid chromatography – organic carbon detection (LC-OCD) system was also used to characterise the DOM pool, including compounds not captured by the analytical window of FT-ICR-MS. With size-exclusion chromatography coupled to a Gräntzel thin-film UV reactor, this procedure differentiated[36]: (i) non-humic "high-molecular-weight substances" (HMWS) of hydrophilic character; (ii) aromatic humic or "humic-like substances" (HS) with higher aromaticity based on UV absorbance at 254 nm; and (iii) "low-molecular-weight substances" (LMWS) including acidic and neutral substances. DOC fractions and molecular weights were determined based on compound retention times using the dominating HS peak in the carbon detector chromatograms to identify the position of other fractions. The position of the HS peak and molecular weight of the HS fraction were calibrated using International Humic Substance Society Suwannee River humic acid and fulvic acid standards[36].

## Soil sampling

Microbial activity was measured in soils, as pore sizes of the soil-water sampling devices may have excluded soil-bound microbes contributing to the DOM pool. Soils were collected using manual shovelling at each hillslope position in two catchments. Three sterilised 30 cm PVC cores were hammered into an exposed soil wall at each sampling depth (5, 15, 30, and 60 cm). Soil from these cores was homogenised and sieved through a 6 mm mesh to exclude roots and inorganic material using sterile tools. A 10 g subset of soil was cooled on ice for bacterial productivity and respiration experiments performed within 24 h of sampling. The remaining samples for bacterial and fungal taxonomic diversity (250 mg) and CAZyme gene quantification and identification (2 g) were stored in sterile centrifuge tubes or freezer bags and frozen on-site using dry ice in ethanol. Samples used for sequencing and microbial activity (enzyme assays and substrate use) were stored at −80 °C and −20 °C, respectively.

## Environmental predictors of DOM

We measured 475 environmental variables in soils and their pore-waters to partition variation in DOM composition. The variables were associated with microbial metabolic activity and biomass (n = 38), microbial diversity (n = 6), expression of CAZymes (n = 412), and soil-water chemical and physical properties (i.e., physico-chemical conditions: n = 19) (see Table S8 for full list).

We measured 38 variables related to microbial activity and biomass. Bacterial production (BP) was measured using $^3$H leucine incorporation after Bååth et al.[66,67] (Supplementary Methods). Decays per minute measured were converted to BP (mg $L^{-1}$ day$^{-1}$) using standard conversion factors[68]. For microbial basal respiration, 1 g of field-moist soil was placed in a 10 mL glass vial and incubated for 24 h in the dark at room temperature (21 °C) without manipulating moisture levels. Incubations of live cells were initiated within 5 h of soil collection. Respired $CO_2$ (μL min$^{-1}$ g$^{-1}$ dry soil) collected in the headspace of the vial was measured with an infra-red gas analyser (QS102, Qubit Systems, Canada). We also measured the potential activity of four hydrolytic enzymes in cell suspensions: beta-xylosidase and beta-glucosidase that break down xylose and oligosaccharides, respectively, N-acetyl-B-D-glucosaminidase that degrades glycoside and amino sugars, and phosphatase that degrades proteins. All enzyme activities were assayed in 96-well plates under controlled conditions (pH 5, room temperature for 1-h) using 4-methylumbelliferone-fluorescence tagged substrates and measured with a Synergy H1 Hybrid spectrophotometer/fluorometer (BioTek Instruments, USA) as in existing protocols[69,70]. To complement the enzyme assays, we measured microbial substrate use of 31 different carbon sources using Biolog® EcoPlates™ (Biolog Inc., USA). Cell suspensions were prepared by adding 10 g dry soil equivalent to 95 mL of sterile NaCl solution (0.85%), 6 ceramic beads and 5 g glass microspheres. Samples were mixed on an orbital shaker for 30 min at 200 rpm, left to settle for

15 min, and a serial dilution on 1 mL supernatant performed to an end concentration of 1:1000 in saline. Inoculated plates were incubated in the dark at 25 °C, and absorbance read every 24 hours on the Synergy H1 microplate reader. Blank wells were always subtracted to reduce noise. Average well colour development (ACWD) was calculated as the rate of change over a 24 h period from day 1 to day 2, as day 2 produced the greatest ACWD indicating the least chance of substrate limitation. Finally, bacterial biomass was measured using flow cytometry. Bacterial cells were separated from soil matrices using buoyant density centrifugation adapted from ref. 71 (Supplementary Methods). Flow cytometry was performed on Accuri™ C6 Plus flow cytometer (BD Biosciences, UK) equipped with a 200 mW solid-state laser emitting light at 488 nm, measuring green fluorescence at 520 nm (FL1 channel). The FL1 and forward scatter detectors were used to reduce autofluorescence found in environmental samples. We ran samples in triplicate, passing 100 µL per technical replicate through the flow cytometer at a speed of 60 µL min$^{-1}$ to prevent overlap of scatter events. Gates were set by comparing scatter plots produced from stained and unstained samples. Flow cytometer counts were validated with Spherotech 8-peak and 6-peak beads. Bacterial biomass (mg g$^{-1}$ dry soil) was calculated from cell counts, assuming a conversion factor[72] of 58 fg cell$^{-1}$.

Bacterial and fungal taxonomic diversities were assessed using exact sequence variants (ESVs) generated by amplicon sequencing of the 16S rRNA gene and ITS2 region, respectively. DNA was extracted from 250 mg of homogenised soil using the DNeasy PowerSoil Pro kit and the QIAcube® (Qiagen, Germany) automated platform. 16S rRNA and ITS2 libraries were prepared following ref. 73 with well-established primers (Table S11). The one exception was that the first set of PCR reactions were set up by mixing 25 µL of HotStarTaq Plus Master Mix, 19 µL RNase-Free Water (Qiagen, homogenised), 0.5 µL of 10 µM primer and 5 µL of gDNA at 5 ng µL$^{-1}$. Indexed and purified amplicons were quantified using the Synergy™ Mx Microplate Reader (BioTek Instruments, USA) before pooling at equimolar concentration. Libraries were sequenced paired-end (2 × 250 bp) on the Illumina Miseq platform at the Aquatic and Crop Resource Development Research Centre, National Research Council Canada, Saskatoon at an average (±SE) read depth of 23594 (±863) and 34,017 (±2380) reads for 16S and ITS, respectively. Sequence data were processed using the MetaWorks pipeline version 1.4.0[74]. To reduce potential bias introduced by both large differences in read depth (i.e. >10-times difference) and small, uneven libraries, we removed samples with <1000 reads and remaining samples were rarefied to the 15th percentile of reads (7150 and 11,103 for 16S and ITS, respectively) using the rrarefy function in vegan[75]. Eight 16S and six ITS samples with slightly less than the 15th percentile were also kept (≥6097 and ≥7544 reads for 16S and ITS, respectively) based on rarefaction curves that showed saturation. Reads were then taxonomically annotated with the RDP classifier v2.13 and UNITE classifier v2.0 for 16S and ITS, respectively. We calculated bacterial and fungal diversity with the Shannon–Weiner index that accounts for relative abundances of ESVs in addition to their number using the diversity function in the R package vegan[75].

To identify CAZyme genes and quantify their transcripts, we used metagenome and metatranscriptome shotgun sequencing, respectively. Shotgun metagenomic libraries were prepared with the Nextera XT DNA library preparation kit and the Nextera XT Index kit v2 (Illumina, USA) following the manufacturer's instructions using the same input DNA that was used for amplicon sequencing. DNA libraries were purified with Agencourt AMPure XP beads (Beckman Coulter, USA) and fragment size (250–1000 bp) verified on a 2100 Bioanalyzer with a high-sensitivity DNA kit (Agilent, USA). Libraries were quantified with the Qubit BR dsDNA assay kit and pooled at equimolar concentrations prior to pair-end sequencing (2 × 150 bp) at the Centre d'Expertise et de Services Génome Québec on an Illumina Novaseq platform. Metatranscriptomes were obtained by extracting RNA from 2 g of soil using the RNeasy® PowerSoil Total RNA Kit (Qiagen, Germany), except that the phenol/chloroform step was repeated twice. The pellet was suspended in 50 µL RNase/DNase-free water, treated with the RNA Clean & Concentrator-5 with DNase I treatment kit (Zymo Research, USA), and eluted in 15 µL of DNase/RNase-free water. RNA quality was verified with the 2100 Bioanalyzer using the RNA 6000 Nano or Pico assay (Agilent, USA), while RNA concentration was determined with the Qubit RNA high-sensitivity assay kit (Life Technologies, USA). Absence of residual DNA in RNA extracts was further confirmed by PCR amplification of the 16S gene. rRNA was depleted from RNA extracts using the Pan-Prokaryote riboPOOL-kit (siTOOLs Biotech, Germany) with hydrophilic streptavidin magnetic beads (New England Biolabs, USA). rRNA-depleted RNA was then purified with the RNA Clean & Concentrator-5 kit and eluted into 10 µL of DNase/RNase-free water. Libraries were prepared using the NEBNext Ultra™ II RNA Library Prep Kit for Illumina (New England Biolabs, USA) and the NEBNext Multiplex Oligos for Illumina kit (New England Biolabs, USA) following the manufacturer's protocol for rRNA-depleted RNA. A quality check of the libraries was performed on the 2100 Bioanalyzer with the high-sensitivity DNA kit (Life Technologies, USA) prior to pooling and pair-end sequencing (2 × 125 bp) on an Illumina HiSeq platform at the Aquatic and Crop Resource Development Research Centre, National Research Council Canada, Saskatoon. Metagenomes were screened with Fastp[76] for read adaptor removal and co-assembled per sampling site with metaSpades[77] (v0.6.1) using Kbase[78] according to default parameters and including the BayesHammer option for read error correction[79]. Gene sequences were identified on the assembled contigs using Prodigal[80] and then annotated as CAZymes using Hidden Markov Models from dbCAN (v9[81], e-value < 1e−15; coverage > 0.35) in local searches with HMMER v3.1b1[82]. Metatranscriptomes were quality-filtered with Fastp according to default parameters[76] and mapped against gene sequences confirmed as CAZymes to obtain their expression profiles using CoverM (v0.6.1 using the 'tpm' option, https://github.com/wwood/CoverM). Transcript counts were normalised using the R package DESeq2 to correct for library size and composition and allow for comparison between samples[83].

Finally, major ions, nutrients, and metal concentrations were measured from a subset of the soil pore water at the Great Lakes Forestry Centre, Sault Ste. Marie, Ontario, according to methods outlined in Table S12. Soil moisture content was directly measured by weighing the change in the mass of ca. 10 g of soil before and after drying in an oven for 24 h at 105 °C degrees relative to the original mass.

## Statistical analyses

We tested if the molecular composition of DOM varied with soil depth and hillslope position using generalised linear models in R version 4.1.2. The probabilities of detecting universal molecules, percent relative abundance of universal molecules, and intensity-weighted mass-to-charge ratios ($m/z_{wm}$) were predicted at each soil depth and hillslope position, while also allowing responses to vary simply because of catchment identity and if the sample was soil or stream. We also allowed depth and position to interact and dropped this term where it was not statistically significant. We used a binomial error structure for models where proportions of counts were predicted and accounted for over-dispersion by including an observation-level random effect. Models were otherwise fitted with a Gaussian error structure and responses that were a proportion of continuous variables were log- or logit-transformed to normalise residuals. Marginal means were predicted at each depth and position by averaging across catchments and either depths or positions using the R package emmeans[84].

To compare the importance of environmental and spatial drivers underlying DOM composition, we used a redundancy analysis (RDA) inferential framework[85]. We partitioned the total unique and shared

variation in the relative abundance of each compound class explained by all environmental variables, spatial structure, and the depth and hillslope gradients using RDA ordination estimated with the varpart function from the vegan package[86]. Spatial structure was modelled using principal coordinates of neighbour matrices[87]; see Supplementary Methods. Prior to the variance partition analysis, the 475 environmental predictors were reduced to avoid overfitting. Where two or more variables were correlated with a Pearson $|r| > 0.80$, we removed the variable with the largest mean $|r|$ with all other variables. We further reduced this subset ($n = 364$) for each compound class with stepwise model selection[88] using the ordistep function from the vegan package, dropping variables that were weakly associated ($p > 0.10$) with molecular composition. The importance of individual environmental predictors in each compound class were tested using hierarchical partitioning generalised to multiple predictor matrices implemented with the R package rdacca.hp[89]. All environmental predictors were scaled to zero mean and unit variance.

To test differential expression of genes annotated as CAZymes, we fitted mixed effects models to raw transcript counts with negative binomial error structures using the R package glmmSeq[90]. Models were fitted to each CAZyme individually with soil depth, hillslope position, and catchment identity as predictors. To account for variation from repeated soil core measurements, we added core identity as a random effect. The dispersion parameter for the negative binomial distribution was estimated separately for each gene with the R package edgeR[91]. We used Wald $\chi2$ tests to test the statistical significance of predictors, and $p$ values were adjusted for multiple comparisons with a false discovery rate ($q$-value) <0.05 considered statistically significant[90]. To visualise differences in expression of CAZymes identified as important in the variance partition analysis, we used PCA. We tested if depth and hillslope explained variation in the first two axes of the PCA using a permutation test with the envfit function from vegan[75]. Permutations to assess the statistical significance for depth and hillslope were stratified within core identity and catchment identity, respectively.

### Reporting summary
Further information on research design is available in the Nature Portfolio Reporting Summary linked to this article.

## Data availability
DNA and RNA sequence data generated in this study have been deposited in the Sequence Read Archive under BioProject accession number PRJNA858277 Lysimeter coordinates, FT-ICR-MS data, compound calculations, CAZY assignment metadata, and amplicon read count summary used in this study have been deposited in Zenodo https://doi.org/10.5281/zenodo.8109398.

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

## Acknowledgements

We are thankful to K Waxenberg, W Pallier, and L Mahony (Cambridge) for help with fieldwork. Additional support was provided by K. Klapproth and I Ulber with FT-ICR-MS measurements and C Schmalsch with LC-OCD samples. We also thank staff at the Natural Resources Canada – Canadian Forest Service (NRCan-CFS) for considerable support, including with logistics (K Webster, P Hazlett), development of extraction and assay protocols (V Rouleau, M-J Morency), conducting laboratory extractions and assays (E Smenderovac, D Chartrand, M-J Morency), and general field and laboratory assistance (J Schadenberg and many others). This work was funded by a Gates Cambridge Scholarship (OPP1144) awarded to E.C.F., NRCan-CFS Cumulative Effects programme awarded to E.J.S.E., H2020 ERC Grant FLUFLUX (716196) to G.S., H2020 ERC Grant sEEIngDOM (804673) to A.J.T., and grant from the Genomic Research and Development Initiative of the Government of Canada to C.M. and E.J.S.E.

## Author contributions

E.C.F. designed the study, with contributions from E.J.S.E. and A.J.T. Data collection was carried out by E.C.F. The lab work was performed or supervised by E.C.F., C.E.E., T.G., G.S., T.D. and C.M. Data processing was done by E.C.F., L.P.P.B., and C.E.E. E.C.F. conducted the data analysis, with contributions from A.J.T. The initial draft of the manuscript was written by E.C.F., with contributions from A.J.T. All co-authors reviewed, commented on, and approved the final manuscript.

## Competing interests

The authors declare no competing interests.
