## [Peer Review File · Nature Communications]

Universal microbial reworking of dissolved organic matter along environmental gradientsReviewer #1 (Remarks to the Author):

This manuscript makes a significant contribution to help unlock the "blackbox" dissolved organic matter biogeochemistry. The novel experimental design examining the source of organic matter via soilwater leading into the headwaters allows for the examination of metabolic processes that lead to homogenization of DOM before it reaches the stream. The combination of DOM chemistry, environmental parameters and microbial metabolomics contribute a clearer picture of how DOM is processed in the River Continuum.

The manuscript is well-written and well-edited. A few minor comments on wording can be found below:

Page 6 line 8; awkward sentence, perhaps use another word besides "differently"

Page 8 line 13; "such as because" reads awkwardly, please rephrase.

Reviewer #2 (Remarks to the Author):

In this manuscript, Freeman et al compare the DOM pool along both soil depth and sloping hillside to determine if different DOM pools eventually converge to a common "universal" set of compounds as a consequence of microbial transformation. They use a variety of analytical, enzymatic and sequence-based tools to interrogate these interesting questions about the "degradation cascade".

--- General ---

A major question/concern involves the use of the ICBM-OCEAN database for compound identification. The ICBM-OCEAN publication is behind a paywall that I can not access, so I can't tell if it is targeting oceanic/aquatic compounds, or if it is just a pithy acronym. Knowing this could clear up some of the differences you see with reference 12 - e.g. do you only see 13% similarity in your samples because you may have many soil-specific compounds shared between your two sites that are not in the ICBM-OCEAN database, and reference 12 sees a much higher correlation because they are looking at aquatic systems? I'm just looking for some clarification on the "usefulness" of the ICBM-OCEAN database for soil (and soil water) compounds.

Page 5 - Is your method quantitative? Is "13%" meaning 13% of the 9327 compounds, or 13% of the total molecular composition that you can measure (page 5)? This question is for other percentages listed in this section, unless they specifically say "relative abundance" (like page 8).

Page 8 - For the paragraph beginning with "Convergence towards...", are you suggesting lignin-compounds are increasing in only relative abundance because other compounds are being preferentially degraded, or that microbial activity is actually making lignin-like compounds, or both?

Page 13, paragraph 1 - The "475 measured environmental variabls" are confusing as to what "measurements" you actually have. Are these a combination of both gene expression values, as well as enzymatic activity (basically, the y-axis of Figure 4b)?

Page 13, paragraph 2 - It isn't clear what "functional composition" is referring to. Is this based on the functional annotations from the metagenomes (CAZymes)? Is this from the expression/transcriptomes of these or other genes?

--- Methods ---

Page 23 - How were the soil samples prepared/stored to preserve RNA? How as the RNA extracted?

Page 23 - It isn't clear what the n=6 microbial diversity measurements and n=19 chemical/physical properties are.

Page 24 - What was used for the hydrolytic assays... the cell suspensions? Live cells during the basal respiration assay?

Page 25 - What primers were used for amplification? Need the "metaworks" pipeline reference or website.

Page 25 - The rarefaction sentence (starting with "samples were rarefied after ref 65") is confusing and (as written) doesn't seem to be what ref 65 did. As written, it looks like you rarefied to 7,150 reads, but allowed samples with at least 715 reads to remain?

Page 27 - Need version numbers for metaSpades, dbCAN and coverM. For coverM, which metric did you use (TPM, RPKM, etc..)? For dbCAN did you use the webserver, or did you use the HMMs in local searches? Either way, need the dbcan reference, and the HMMer or other search method reference and version if used locally.

--- Figures ---

Figure 1 - Trying to match the text with the figure, I don't understand many of the percentages listed. The text says Figure 1e had a mean increase of 9.4%, while 1f had an increase 8.1%, but I don't see that if I am meant to be looking at the black boxes/lines.

Figure 2 - Only the bold lines have error bars.

Figure 3 - Like figure 1, many of the percentages in the text are not apparent in this figure. For example, how was there a 68% drop in DOC for the hillslope when it only starts at ~20%? Or, are you not reporting percentages as they are on the y-axis, and instead are reporting the percentage as a ratio drop between the start and end points? Figure 3C for the hill doesn't suggest that, as the text says there is a 7.1% decrease, yet it looks like it goes from ~5 to ~2. Additionally, Figure 3 b,c,d for depth don't seem to be referenced in the manuscript text.

Reviewer #3 (Remarks to the Author):

General Comments

This paper is trying to determine how dissolved organic matter (DOM) changes along soil depths and hillslopes. The paper had a nice field design and looked at four replicate catchments in Ontario, Canada, from hillslope to stream, where porewater was sampled down to 60cm depth and DOM was characterized with FT-ICR-MS.

While this is a very interesting study and integrates a large number of neat datasets, I felt the weight of its claims were not backed up with strong evidence, or its claims were stated in such a way that they come across as counterintuitive. The claims potentially need to be tempered. My perspective as a reviewer is a soil microbial ecologist who is familiar with the decomposition (e.g., cazyms, transcriptomics) and soil biogeochemistry. I needed to read the paper that preceded this paper (Rath et al. 2019) to understand some of the claims, and I think the authors need to spend more time describing some of their assumptions and the literature that supports it. Some of the results show what appears to be very small changes, and the importance of these results appears to be overstated. Other interpretations of the data did not make sense to me, and I request the authors to justify their explanation. I describe these instances in detail below.

Line numbers not provided, so I am providing the page number and line on that page.

Specific Comments

Page 2, Line 2: Accelerate climate change how? Vague

Page 2, Line 14: Need to define what you mean by "universal," the meaning is not clear for a general audience

Page 4, Lines 6-7: I think you need to better set up this premise and provide more detail for the decomposition and microbial assimilation process, otherwise this sounds counterintuitive (i.e. plants tissues are known for their large macromolecular structures, not small molecular weight cmpds, except for some root exudates). For instance, Rath et al. 2019 describes how the low molecular weight plant compounds were generated: "plant-derived molecules were first broken down into molecules containing a large proportion of low-molecular-mass compounds." It would also be helpful if you could provide examples of small plant-derived molecules and large microbial product.

Page 5, Lines 2-3: Which underlying processes? Vague

Page 5, Line 4: Sentence starting with "DOM" is missing something. Is this your hypothesis? If so, should start with something like "we expected..." or "we hypothesized" to signal that

Page 5, Line 8: Why should microbial processing be relatively less important?

Page 5, Lines 8-10: Are these predictions or results? Would be helpful to start with something that signals which... e.g., "we found..." or "we predict that..."

Page 5, Lines 10-14: This is a confusing sentence; work on the number agreement. Perhaps rephrase "Our results implicate that common metabolic processes shift DOM towards..."

Page 5, Lines 17-18: Are these comparable methods? Meaning, would one method yield more compounds than the other?

Page 5, Line 21: Write out ICR at first mention

Page 5, Lines 22-23: Why the difference in percentages? Are the detected compounds the same between studies? If your method detected more rare compounds than the previous study, and the previous study was biased to more abundant compounds, it would make sense that your percentages are lower. If you can subset your compounds to those detected in the previous study, then you can see if your percentages are comparable

Page 5, Line 24: What percentage were unknown?

Page 6, Line 4: Where are the results for the size exclusion chromatography?

Page 6, Line 8: I understand you're using "universal" because of the wording of a previous study, but it is a confusing term in this context. This seems more akin to a "core" microbiome. The terms core/ubiquitous are more understandable to a broad audience

Page 6, Line 9: More detail would be useful to describe your result. Throughout the soil depth profile, at all landscape positions?

Page 6, Line 10: Report statistical test results

Page 6, Line 11: There was no change in what? In porewater? Pool size? Vague

Page 6, Line 11: Looking at this graph, there is a lot of variance, and it is not clear to me that there was no change from along the hillslope gradient. What is the statistical test? Report

Figure 1C-F: What is the statistical test, and do your p values account for multiple comparisons? A Tukey Post-Hoc test could be appropriate here. The stars are halfway off the graphs and are easy to miss.

Page 6, Line 13: It doesn't make sense to start this sentence with "however"; isn't this an addition that agrees with the previous result?

Page 6, Line 13-15: I'm lost...generally in this paragraph, I'm finding it hard to corroborate the written results with the figure, and the results are hard to follow. Please improve the reporting of the results.

Page 6, Line 16: Are these percent differences? Also I'm confused where these numbers came from...when I look at the Fig 1e, the range of percentages is less than 5%.

Page 6, Lines 17-21: This doesn't follow and seems out of place

Page 7, Line 1: Why would a degradation product be larger? Do you mean they were assimilated by microbes (i.e., microbial biomass products)

Page 7, Lines 1-7: These increases are very subtle and do not provide the weight of evidence to claim that this is evidence of progressive re-working of DOM. These are increases of 10 Daltons or a quote "much stronger" increase of 15 Daltons, and are all in the 415-452 Da range. Even if this is statistically significant, it is quite small and I cannot see taking such a strong biological interpretation from this result. The Roth paper that is cited defines low molecular mass as 150-300 Da, mid molecular mass as 300-450Da, and high molecular mass as >450Da. Perhaps there is

signal that is lost in taking the average of all the molecules, which would be seen if you instead looked at how the abundance of these different size classes changed over the hillslope and depth gradients?

Page 7, Line 2: Using molecular weight definitions like this seems coarse and imprecise, given that plant and microbial biomolecules have compounds in these weight ranges.

Page 7, Line 12: n=1216...what? Compounds?

Page 9, Line 13: How many replicates for the means? 4?

Page 8, Line 2: So is this the same data, just aggregated differently? If so, this was a bit confusing

Page 8, Line 5: Specify the statistical test

Page 8, Line 7: This statement is very strong and I am not seeing enough evidence to justify it

Page 8, Lines 7-10: I am confused how an *increase* in lignin-like material would indicate an *increase* in microbial reworking. Lignin is a plant biomolecule. I would interpret an increase of lignin-like compounds with depth as a *decrease* in microbial reworking...meaning the plant material is not being degraded and sticking around longer.

Page 8, Lines 9-11: This result doesn't appear to be the trend seen in Fig 2, why?

Page 8, Lines 13-16: This sentence is confusing. I'm guessing the authors are trying to explain the phenomenon of how changes in relative abundance can be caused by other compounds going down, but it's confusing

Page 8, Line 20: Please report statistical significance, and provide a figure for these results (either in text or in the supplemental). I initially was very confused because I thought this data was in Fig.2 and the results don't match—because Fig 2 is not showing abundance, it's showing changes in molecular mass. Generally, I think it would be useful to see the same graphs but representing the abundance of the compounds.

Page 9, Line 1: Does the interpretation of increasing molecular mass apply for a molecular category that you're attributing primarily to plants? If microbes have degraded or depolymerized the unsaturated hydrocarbons, they would make smaller unsaturated hydrocarbons, not larger unsaturated hydrocarbons (because microbes aren't generating unsaturated hydrocarbons). The "increasing molecular mass" hypothesis makes sense to me when a microbe converts a low molecular weight plant exudate (e.g., monosaccharides) and assimilates it into biomass (e.g., proteins, oligosaccharides), but not when trying to explain the increase in molecular weight of plant-derived molecules.

Page 9, Line 3: Changed how? Vague, please specify (they increased)

Page 9, Lines 13-15: The authors are basing a lot of interpretation off this paper. As I indicate in the comments above, I am not seeing how the evidence shown in this paper justifies this conclusion.

Figure 2: The colors in the legend do not match the image because it appears transparency was applied to the lines (please make them match). "Darkened" lines would be better written as "solid" lines, and the rest are dashed

Page 11, Lines 2-4: The decrease of C concentrations with depth does not necessarily reflect continuous microbial degradation, it reflects that there are lower amounts of C at depth. This is generally true of most soil depth profiles in mineral soils—soil TOC decreases with depth, and that C is typically older than the surface C. It is more likely that DOC concentrations are correlated with TOC concentrations.

Page 11, Lines 4-5: Also, this is a very normal thing to see in soil profiles—microbial activity and abundance decreases with depth as TOC concentrations decrease. You could cite some papers and do an "as expected" part here

Page 11, Lines 6-7: Why isn't the size-exclusion chromatography data presented in the main paper, and how are you defining plant- vs microbial-derived C?

Page 11, Lines 12-15: I would reverse the order of these two pieces of information; present the result first (decline in proteins), followed by interpretation.

Page 11, Lines 16-17: I am seeing no evidence here of a degradation cascade, let alone two degradation cascades. I see some very normal patterns for soil depth profiles taken from mineral soils. And no metabolic pathways are discussed. Please revise and provide a more tempered conclusion that reflects the data presented.

Figure: 3: How do you have C:N values close to zero? Why not show these as regressions and provide r-squared, etc? And finally, what is the statistical test, and do your p values account for multiple comparisons? If not, a Tukey post-hoc test would be more appropriate.

Page 13, Line 4: What are these environmental variables? Are these different molecule categories, in addition to edaphic factors like pH etc? This is a large number of variables

Page 13, Line 6-7: See comment for "Page 9, Line 1"; if you're attributing unsaturated hydrocarbons primarily to plants, it does not make sense that microbes would make them bigger when they degrade them; they would become smaller as they get depolymerized (because microbes don't generate them, they only degrade them). And for zones where plant root exudation is low (e.g., 60cm), overall the whole system is more microbial because plant input is low...why would increased carbohydrate molecular weight be interpreted as more degradation in this case?

Page 13, Line 11: I don't understand what you mean by "it was comparable on a per-variable-basis" as a justification for them still being important even though they're only explaining <4% of the variation...

Page 13, Line 13: What variables explained the most variation?

Page 13, Line 19: This is a lot of variables! Are these variables edaphic factors or molecular classes?

Page 14, entire paragraph: I would like the authors to provide citations for the activities of these enzyme classes. Also, many of these enzyme classes contain enzymes that can act of many types of substrates, and can have plant/fungal/or bacterial origin. I would like more support for their assertions.

Page 14, Line 1: these CAZymes can also potentially break down microbial carbohydrates

Page 14, Line 5: I'd change "these genes" to "these CAZymes" to make it clear you're specifying the CAZymes from the previous sentence. Or you could use a semicolon, but then the sentence is very long

Page 14, Line 10: What do you mean by "at a community-level"? Are you talking about bulk measurements from enzyme activity assays?

Figure 4: Fig 4b "activity" misspelled. I'm a bit confused about the different sets of totals—please clarify your presentation. Do all the totals add up to 100%? For instance, for the Unsat Hydrocarbons, there is a total variation explained by model (0.66...66%?), and Envi Variation (maybe 95%?), 4% spatial variation, and then Gene Activity (66%), Enzyme activity (25%) and Physical Chemistry (8%). I'm guessing that everything in Fig 4b represents the Envi Variation? (so it's actually like...of the 66% of total variation explained by the model, Gene activity represents 43% of the variation (66% of the total 66% of variation explained by the model).

Page 16, Lines 9-10: Alright, I'm not seeing this. What data showed that you switched from plant processing genes to microbial processing genes? Also you don't show any gene expression across the gradients; you're showing what % of the variation in the model for a particular class of compounds is explained by different CAZymes.

Page 16, Line 9: What is the justification for using "universally" here?

Page 17, Line 5: I agree that the data suggests that microbes are processing compounds of microbial origin rather than plant origin at depth. However, the phrase of "transforming plant-derived compounds into higher molecular weight [compounds]" makes it sounds like there are plant inputs down there, and that is not what your data suggests. That messaging is very confusing. At depth, the source is microbial. I suggest you rephrase.

Page 17, Line 10: What is the evidence for removing the source-specific components?

Fig 5: I don't understand the y-axis. Is this percent similarity to deep sea samples? If so, change the unit from (%) to (% similarity).

Page 17, Line 20: I would flesh this out a little bit more (e.g., deep soils and toe/stream were most similar to deep sea). Though the error bars are pretty large...and surface soil is still very similar, so it increases but only from 39% to maybe 43%, not accounting for error bars. The Hillslope position looks clearer. As I asked before, is this the same data presented twice, but just aggregated differently? (by either hillslope or depth) That should be explained

Page 19, Line 5: How does persistence follow here?

Page 19, Line 8: Which were the non-universal? Reminder would be helpful

Page 19, Line 13: Previously you were saying (and I think correctly), that pectate lyases are associated with plant material and decreased over the soil profiles. It doesn't make sense for it to be associated with general degradation processes

Page 19, Lines 14-15: You did not present data on the taxonomic composition of pectate lyases, but you appear to be invoking functional redundancy here. You cannot say that pectate lyase mediated degradation is independent of microbial taxonomic composition unless you test this.

Page 19, Lines 17-18: What consistent patterns are you citing? Be more specific. It sounds like you're talking about CAZyme data; if so, what consistent patterns within that?

Page 19, Line 22: Change "individuals" to "individual"

Page 19, Line 23: Vague, what about your methods would allow you to test this?

Page 19, Line 24: What have you presented that would help you estimate the degradation state?

REVIEWER COMMENTSReviewer #1 (Remarks to the Author):

This manuscript makes a significant contribution to help unlock the “blackbox” dissolved organic matter biogeochemistry. The novel experimental design examining the source of organic matter via soilwater leading into the headwaters allows for the examination of metabolic processes that lead to homogenization of DOM before it reaches the stream. The combination of DOM chemistry, environmental parameters and microbial metabolomics contribute a clearer picture of how DOM is processed in the River Continuum. The manuscript is well-written and well-edited. **We thank the Reviewer for this positive feedback.**

A few minor comments on wording can be found below:

Page 6 line 8; awkward sentence, perhaps use another word besides “differently”

We have rephrased this sentence so that the use of “differently” is clearer and less awkward.

Page 8 line 13; “such as because” reads awkwardly, please rephrase.

Rephrased to “for example, if”.

Reviewer #2 (Remarks to the Author):

--- General ---

R2.1. A major question/concern involves the use of the ICBM-OCEAN database for compound identification. The ICBM-OCEAN publication is behind a paywall that I can not access, so I can't tell if it is targeting oceanic/aquatic compounds, or if it is just a pithy acronym. Knowing this could clear up some of the differences you see with reference 12 - e.g. do you only see 13% similarity in your samples because you may have many soil-specific compounds shared between your two sites that are not in the ICBM-OCEAN database, and reference 12 sees a much higher correlation because they are looking at aquatic systems? I'm just looking for some clarification on the "usefulness" of the ICBM-OCEAN database for soil (and soil water) compounds.

We apologise for the confusion. ICBM-OCEAN is a tool to assign molecular formulae to mass spectrometry data. ICBM-OCEAN is not a database of formulae. OCEAN does sound marine-centred but is simply part of an acronym (Oldenburg–Complex Molecular Mixtures, Evaluation & Analysis). Its applicability to a wide range of sample types has been proven in many studies (cited 57 times). In the original publication, the authors also state that the tool has been tested on marine and fresh waters and even microbial cultures (page 6833, at the end of paragraph 2 after Figure 1). Though the paper is behind a paywall, the tool is freely accessible at: <https://rhea.icbm.uni-oldenburg.de/geomol/>. The reader can recreate our analyses with the above link and the provided data (10.5281/zenodo.8109398.). We have now replaced “platform” with “formula assignment and analysis tool” and added a link that is not behind any paywall.

Changed on lines 467-470:

“Masses ranging from 150 to 1000 m/z were exported from the Bruker Data Analysis software, and we assigned molecular formulas using the online formula assignment and analysis tool ICBM-OCEAN³⁴ (freely available at <https://rhea.icbm.uni-oldenburg.de/geomol/>).”

R2.2. Page 5 – Is your method quantitative? Is “13%” meaning 13% of the 9327 compounds, or 13% of the total molecular composition that you can measure (page 5)? This question is for other percentages listed in this section, unless they specifically say “relative abundance” (like page 8).

Yes, this method (size exclusion chromatography) is entirely quantitative and we have now explained this on line 135:

“Using size-exclusion chromatography to quantify DOM fractions...”

13% is the number of samples that contain HMWS. The remaining percentages refer to the relative count of molecules in a specified class based on FT-ICR-MS. To clarify, we added “based on counts of molecular formulae” on line 132.

Further changes were made on lines 136-138:

“...HMWS were detected in only 13% of soil pore water samples and, when found, contributed, on average (95% confidence interval, CI), only 8% (3 to 12%) to the total dissolved organic carbon concentration (Table S1).”

R2.3. Page 8 – For the paragraph beginning with “Convergence towards...”, are you suggesting lignin-compounds are increasing in only relative abundance because other compounds are being preferentially degraded, or that microbial activity is actually making lignin-like compounds, or both?

We have altered the text on lines 175-176 as follows:

“Convergence towards a universal DOM pool was due to the preferential degradation of subsets of compounds actively reworked by microbes.”

R2.4. Page 13, paragraph 1 - The "475 measured environmental variabls" are confusing as to what "measurements" you actually have. Are these a combination of both gene expression values, as well as enzymatic activity (basically, the y-axis of Figure 4b)?

Added on lines 273-275:

“We measured 475 environmental variables related to the gene activity of carbohydrate-degrading enzymes, the activity of extracellular enzymes or level of carbohydrate substrate utilisation, and soil physical chemistry (Table S8).”

The full list of variables is now given in a new Supplementary Table 8.

R2.5. Page 13, paragraph 2 - It isn't clear what "functional composition" is referring to. Is this based on the functional annotations from the metagenomes (CAZymes)? Is this from the expression/transcriptomes of these or other genes?

We are referring to both functional annotations (so-called potential function) and expression/transcriptomes.

Clarified as follows on lines 285-287:

“Most of the environmental variation in DOM composition was due to the potential and realised activity of microbial communities, particularly for compound classes that reflected increased processing along the degradation cascades.”

--- Methods ---

R2.6. Page 23 - How were the soil samples prepared/stored to preserve RNA? How as the RNA extracted?

We added the following on lines 500-501:

“The remaining samples for bacterial and fungal taxonomic diversity (250mg) and CAZyme gene quantification and identification (2g) were stored in sterile centrifuge tubes or freezer bags and frozen on-site using dry ice in ethanol.”

Extraction details are on lines 579-581:

“Metatranscriptomes were obtained by extracting RNA from 2 g of soil using the RNeasy® PowerSoil Total RNA Kit (Qiagen, Germany), except that the phenol/chloroform step was repeated twice.”

R2.7. Page 23 - It isn't clear what the n=6 microbial diversity measurements and n=19 chemical/physical properties are.

We have added Table S8 to clarify which variables were analysed.

Changed on lines 517-520:

“The variables were associated with microbial metabolic activity and biomass (n=38), microbial diversity (n=6), expression of CAZymes (n=412), and soil water chemical and physical properties (i.e., physico-chemical conditions: n=19) (see Table S8 for full list).”

R2.8. Page 24 - What was used for the hydrolytic assays... the cell suspensions? Live cells during the basal respiration assay?

Clarified on lines 518-519:

“We also measured the potential activity of four hydrolytic enzymes in cell suspensions: ...”

Live cells were used for the basal respiration assay. We have clarified this on lines 516-517: “Incubations of live cells were initiated within 5 hours of soil collection.”

R2.9. Page 25 - What primers were used for amplification? Need the "metaworks" pipeline reference or website.

Primers have been added to Table S12 now cited on line 551 of the Main Text.

We have also added the following reference to the MetaWorks pipeline:

Porter, T. M., & Hajibabaei, M. (2022). MetaWorks: A flexible, scalable bioinformatic pipeline for high-throughput multi-marker biodiversity assessments. *PLOS ONE*, 17(9), e0274260. doi: 10.1371/journal.pone.0274260

R2. 10. Page 25 - The rarefaction sentence (starting with "samples were rarefied after ref 65") is confusing and (as written) doesn't seem to be what ref 65 did. As written, it looks like you rarefied to 7,150 reads, but allowed samples with at least 715 reads to remain?

Ref 65 was cited to highlight rarefying as a normalization technique to address variable library size and the importance of removing samples with small (<1000) and uneven libraries (> 10× difference). Samples with less than 1000 reads were removed in our case. We also only calculated alpha diversity from the amplicon data in our analyses, which consider relative abundance. There was no differential abundance testing with these amplicon data.

Changed on lines 560-566:

“To reduce potential bias introduced by both large differences in read depth (i.e. >10-times difference) and small, uneven libraries, we removed samples with <1000 reads and remaining samples were rarefied to the 15th percentile of reads (7150 and 11103 for 16S and ITS, respectively) using the rrarefy function in vegan^{65,66}. Eight 16S and six ITS samples with slightly less than the 15th percentile were also kept (≥6097 and ≥7544 reads for 16S and ITS, respectively) based on rarefaction curves that showed saturation.”

R2.11. Page 27 - Need version numbers for metaSpades, dbCAN and coverM. For coverM, which metric did you use (TPM, RPKM, etc.)? For dbCAN did you use the webserver, or did

you use the HMMs in local searches? Either way, need the dbcan reference, and the HMMer or other search method reference and version if used locally.

Thank you for helping us ensure that the analysis is maximally reproducible. We have added the requested details on lines 598-606:

“Metagenomes were screened with Fastp⁷⁵ for read adaptor removal and co-assembled per sampling site with metaSpades⁷⁶ (v0.6.1) using Kbase⁷⁷ according to default parameters and including the BayesHammer option for read error correction⁷⁸. Gene sequences were identified on the assembled contigs using Prodigal⁷⁹ and then annotated as CAZymes using Hidden Markov Models from dbCAN (v9⁸⁰, e-value < 1e-15; coverage > 0.35) in local searches with HMMER v3.1b1⁸¹. Metatranscriptomes were quality-filtered with Fastp according to default parameters⁷⁵ and mapped against gene sequences confirmed as CAZymes to obtain their expression profiles using CoverM (v0.6.1 using the ‘tpm’ option, <https://github.com/wwood/CoverM>).”

--- Figures ---

R2.12 - Figure 1 - Trying to match the text with the figure, I don't understand many of the percentages listed. The text says Figure 1e had a mean increase of 9.4%, while 1f had an increase 8.1%, but I don't see that if I am meant to be looking at the black boxes/lines.

We presented the relative not absolute change. We have clarified the text to align more clearly with the figure on lines 148-152:

“The relative abundance of universal compounds increased by (95% CI) 9.4% (5.9 to 12.9%, linear model: $t=5.5$, $p < 0.001$, $df=57$) from an estimated mean of 54.3% (51.8 to 56.8%) at 5 cm to 64% (61.4 to 66.0%) at 60 cm (Fig. 1e). There was a similar 8.1% (6.4 to 9.7%, $t=3.0$, $p=0.029$, $df=57$) relative increase in universal compounds from 56.9% (54.4 to 59.3%) at the shoulder to 59.8% (54.9 to 64.5%) at the stream, respectively (Fig. 1f).”

R2.13- Figure 2 - Only the bold lines have error bars.

We did not mean to imply that non-bolded lines have no errors. These errors are reported in the supplement. Instead, we wanted to simplify the visual information. We have included the plot with error bars to demonstrate how messy it is with the error bars included below. To make our choice clearer we have added the following text on lines 231-232:

“Errors for non-statistically-significant compound classes are presented in Table S5”

Plot with error bars:

R2.14- Figure 3 - Like figure 1, many of the percentages in the text are not apparent in this figure. For example, how was there a 68% drop in DOC for the hillslope when it only starts at ~20%? Or, are you not reporting percentages as they are on the y-axis, and instead are reporting the percentage as a ratio drop between the start and end points? Figure 3C for the hill doesn't suggest that, as the text says there is a 7.1% decrease, yet it looks like it goes from ~5 to ~2. Additionally, Figure 3 b,c,d for depth don't seem to be referenced in the manuscript text.

Yes, we are reporting the percentage as a ratio drop between the start and end points, that is, we are reporting the relative change. We have updated the entire manuscript so that the percent change from start to end point is reported alongside the word “relative”. We have also added a reference to Fig. 3c (line 243 and 245) that was missing from the text, whereas references to 3b and 3d are given on line 239 and line 239, respectively.

Reviewer #3 (Remarks to the Author):

This paper is trying to determine how dissolved organic matter (DOM) changes along soil depths and hillslopes. The paper had a nice field design and looked at four replicate catchments in Ontario, Canada, from hillslope to stream, where porewater was sampled down to 60cm depth and DOM was characterized with FT-ICR-MS.

We thank the Reviewer for their positive feedback here and address their concerns below. When comments referred to a similar concern, we have grouped them together and described

our solution below this group. We also want to thank the Reviewer for providing such a thorough and well thought out review.

R3.1. While this is a very interesting study and integrates a large number of neat datasets, I felt the weight of its claims were not backed up with strong evidence, or its claims were stated in such a way that they come across as counterintuitive. The claims potentially need to be tempered. My perspective as a reviewer is a soil microbial ecologist who is familiar with the decomposition (e.g., cazymes, transcriptomics) and soil biogeochemistry. I needed to read the paper that preceded this paper (Rath et al. 2019) to understand some of the claims, and I think the authors need to spend more time describing some of their assumptions and the literature that supports it. Some of the results show what appears to be very small changes, and the importance of these results appears to be overstated.

We are sorry for the confusion that we have caused about the size of the observed effects. We believe that this concern is largely a wording issue and we have tried to clarify relevant text in response to specific points listed below. We have also spent more time describing our assumptions and the literature that supports them in the Introduction (lines 72-81) as explained below.

Other interpretations of the data did not make sense to me, and I request the authors to justify their explanation. I describe these instances in detail below.

Abstract:

R3.2. Page 2, Line 2: Accelerate climate change how? Vague

Changed on lines 24-25:

“Soils are losing increasing amounts of carbon annually to freshwaters as dissolved organic matter (DOM), which, if degraded, can offset their carbon sink capacity.”

R3.3 Page 2, Line 14: Need to define what you mean by “universal,” the meaning is not clear for a general audience

Added on lines 25-27:

“DOM is more susceptible to degradation closer to its source and becomes increasingly dominated by the same (i.e., universal), difficult-to-degrade compounds as degradation proceeds.”

R3.4. Page 4, Lines 6-7: I think you need to better set up this premise and provide more detail for the decomposition and microbial assimilation process, otherwise this sounds counterintuitive (i.e. plants tissues are known for their large macromolecular structures, not small molecular weight cmpds, except for some root exudates). For instance, Rath et al. 2019 describes how the low molecular weight plant compounds were generated: “plant-derived molecules were first broken down into molecules containing a large proportion of low-molecular-mass compounds.” It would also be helpful if you could provide examples of small plant-derived molecules and large microbial product.

We have added a paragraph in the Introduction to clarify this concept. Added on lines 71-77: “ Degradation of DOM is instead characterised by increasing molecular weight. The process generating this increase in weight likely reflects the extracellular microbial decomposition of large macromolecular plant material into smaller molecules, for example, the production of simple unsaturated oligogalacturonates after pectin degradation²⁴. These small metabolites are then transformed to larger microbially-derived compounds, namely complex polysaccharides associated with microbial tissues and products, such as chitin or glucans^{23,25}.”

(Comments moved up from below)

R3.25-26.

Page 7, Line 1: Why would a degradation product be larger? Do you mean they were assimilated by microbes (i.e., microbial biomass products)

Yes, broken down into smaller compounds and then assimilated. However, we have removed this paragraph and so this comment is no longer applicable.

Page 7, Line 2: Using molecular weight definitions like this seems coarse and imprecise, given that plant and microbial biomolecules have compounds in these weight ranges.

We agree and have removed this paragraph.

R3.37. Page 9, Line 1: Does the interpretation of increasing molecular mass apply for a molecular category that you're attributing primarily to plants? If microbes have degraded or depolymerized the unsaturated hydrocarbons, they would make smaller unsaturated hydrocarbons, not larger unsaturated hydrocarbons (because microbes aren't generating unsaturated hydrocarbons). The "increasing molecular mass" hypothesis makes sense to me when a microbe converts a low molecular weight plant exudate (e.g., monosaccharides) and assimilates it into biomass (e.g., proteins, oligosaccharides), but not when trying to explain the increase in molecular weight of plant-derived molecules.

R3.39. Page 9, Lines 13-15: The authors are basing a lot of interpretation off this paper. As I indicate in the comments above, I am not seeing how the evidence shown in this paper justifies this conclusion.

R3.51. Page 13, Line 6-7: See comment for "Page 9, Line 1"; if you're attributing unsaturated hydrocarbons primarily to plants, it does not make sense that microbes would make them bigger when they degrade them; they would become smaller as they get depolymerized (because microbes don't generate them, they only degrade them). And for zones where plant root exudation is low (e.g., 60cm), overall the whole system is more microbial because plant input is low...why would increased carbohydrate molecular weight be interpreted as more degradation in this case?

The capability for hydrocarbon degradation but also synthesis is widespread among microorganisms (both prokaryotes and eukaryotes) (Ladygina et al., 2006). For example, in Koster et al., 1999, the release of hydrocarbons was suggested to be connected with the functioning of cytoplasmic membrane: the hydrocarbons of low molecular weight passed through the membrane into the culture broth, whereas the retained hydrocarbons were elongated. A more recent review is available in Vaishnavi et al., 2021.

We address this group of comments with the following:

1. We add background material to the Introduction (lines 71-81): "Degradation of DOM is instead characterised by increasing molecular weight. The process generating this increase in weight likely reflects the extracellular microbial decomposition of large macromolecular plant material into smaller molecules, for example, the production of simple unsaturated oligogalacturonates after pectin degradation²⁴. These small metabolites are then transformed to larger microbially-derived compounds, namely complex polysaccharides associated with microbial tissues and products, such as chitin or glucans^{23,25}. Many microbes synthesise carbohydrates, hydrocarbons, and lipids in this way²⁷. This process partly reflects the function of semipermeable cell membranes, where diffusion is restricted to molecules of low molecular weight, but these low molecular weight substances are subsequently elongated and incorporated into larger cell structures²⁶."

2. We also refer to the possibility that higher weight compounds are being conserved relative to lower-weight compounds on lines 219-221:
“Together, these results suggested that either larger mass compounds were being conserved or that new, heavier compounds were being created from lighter precursors.”

Koster J, Volkman JK, Rullkotter J, Scholzbottcher BM, Rethmeier J, Fischer U. Monomethyl-branched, dimethyl-branched, and trimethylbranched alkanes in cultures of the filamentous cyanobacterium *Calothrix scopulorum*. *Org Geochem* 1999;30:1367–79.

Ladygina N, Dedyukhina EG, Vainshtein MB. A review on microbial synthesis of hydrocarbons. *Process Biochemistry*. 2006 May 1;41(5):1001-14.

Vaishnavi J, Osborne WJ. Microbial volatiles: small molecules with an important role in intra-and interbacterial genus interactions-quorum sensing. In: *Volatiles and Metabolites of Microbes* 2021 Jan 1 (pp. 35-50). Academic Press-<https://doi.org/10.1016/B978-0-12-824523-1.00005-5>.

R3.5. Page 5, Lines 2-3: Which underlying processes? Vague

We have explicitly mentioned the two broad processes examined in this paper.

Added on lines 102-104:

“...the extent of homogenisation and underlying physico-chemical conditions or microbial processes would differ.”

R3.6. Page 5, Line 4: Sentence starting with “DOM” is missing something. Is this your hypothesis? If so, should start with something like “we expected...” or “we hypothesized” to signal that

Added on lines 105-106:

“We expected DOM sources (i.e., plant litter) to be relatively consistent through ...”

R3.7. Page 5, Line 8: Why should microbial processing be relatively less important?

Thank you for pointing this out. We have clarified as follows:

Added on lines 107-111:

“Along the hillslope, DOM should reflect different sources because of hydrological mixing from different positions that vary in moisture, erosion, vegetation type and rooting depth. Therefore, we expected microbial processing along the hillslope should be relatively less important than with increasing soil depth because of the relatively large variation in DOM sources.”

R3.8. Page 5, Lines 8-10: Are these predictions or results? Would be helpful to start with something that signals which... e.g., “we found...” or “we predict that...”

These are predictions and we modified the text accordingly. Thank you for the suggestion.

Added on lines 111-112:

“In contrast, our second prediction was that DOM.”

R3.9. Page 5, Lines 10-14: This is a confusing sentence; work on the number agreement. Perhaps rephrase “Our results implicate that common metabolic processes shift DOM towards...”

Rephrased exactly as suggested.

R3.10. Page 5, Lines 17-18: Are these comparable methods? Meaning, would one method yield more compounds than the other?

Yes, these are comparable methods – the same method run on the same instrument. As stated in the Roth et al., 2018 paper: “For the FT-ICR-MS measurements extract aliquots were diluted to 20 mg l⁻¹ organic carbon in ultrapure water/methanol (1:1). The Bruker Solarix FT-ICR-MS (15 tesla) at the University of Oldenburg (Germany) was used.”

Added on lines 122-123:

“... more than twice that observed in a previous soil study using the identical FT-ICR-MS method and instrument²³...”

R3.11. Page 5, Line 21: Write out ICR at first mention

We have removed the acronym so this comment is no longer relevant.

R3.12. Page 5, Lines 22-23: Why the difference in percentages? Are the detected compounds the same between studies? If your method detected more rare compounds than the previous study, and the previous study was biased to more abundant compounds, it would make sense that your percentages are lower. If you can subset your compounds to those detected in the previous study, then you can see if your percentages are comparable

As we go on to discuss later, the differences are due to where samples are collected along degradation cascades rather than biases in detection, as the other study again used the same method and instrument. We have clarified this point on lines 127-129:

“Despite the many detected formulae, only 13% occurred in all samples, that is, were universal, compared with between 47 to 87% in a synthesis from headwaters to oceans using the same method and instrument¹².”

R3.13. Page 5, Line 24: What percentage were unknown?

Added on lines 121-123:

“We detected 12,487 peaks and assigned 9327 unique molecular formulae (75% of peaks detected) ...”

R3.14. Page 6, Line 4: Where are the results for the size exclusion chromatography?

We have presented results for the size exclusion chromatography in Fig. 3c. However, we believe future readers may also have this question so have added an additional supplementary Table S1 with the results to address the Reviewer's concern.

R3.15. Page 6, Line 8: I understand you're using “universal” because of the wording of a previous study, but it is a confusing term in this context. This seems more akin to a “core” microbiome. The terms core/ubiquitous are more understandable to a broad audience

We also debated what we should call this group of compounds. Although we want to preserve the link to the original Zark paper also published in Nat Comms, we have added the following text and reference so that a diverse audience can quickly understand our ideas.

Added on line 58:

“they are termed “universal”¹² or “core”¹³”

¹³Stadler, Masumi, et al. "Applying the core-satellite species concept: Characteristics of rare and common riverine dissolved organic matter." *Frontiers in Water* 5: 36.

R3.16. Page 6, Line 9: More detail would be useful to describe your result. Throughout the soil depth profile, at all landscape positions?

Added on lines 139-140:

“Throughout the soil depth profile, the DOM pool converged upon a universal pool but not along landscape hillslope positions (Fig. 1).”

R3.17. Page 6, Line 10: Report statistical test results

Very good point. Thank you. We have corrected here and throughout the manuscript.

Reported on lines 142-143:

... at 60 cm (Fig. 1c, generalised linear model: $z=2.1$, $p=0.036$, $df=57$).

R3.18. Page 6, Line 11: There was no change in what? In porewater? Pool size? Vague

Added on lines 143-145:

“There was no change in the proportion of universal compounds between the shoulder position and streams (Fig. 1d; Table S2), as expected if hydrological mixing was important along the hillslope gradient.”

R3.19-20. Page 6, Line 11: Looking at this graph, there is a lot of variance, and it is not clear to me that there was no change from along the hillslope gradient. What is the statistical test?

Report

Figure 1C-F: What is the statistical test, and do your p values account for multiple comparisons? A Tukey Post-Hoc test could be appropriate here. The stars are halfway off the graphs and are easy to miss.

We fitted generalised linear models and compared marginal means between depths and positions using the emmeans package. This package already adjusts p values with the Tukey method. We now report the statistical tests results throughout the Main Text (see our response to R3.17 above) and added the test information to the figure caption on lines 170-173, along with re-adjusting the position of the stars on the graph so that they are harder to miss:

“Black stars denote hillslope positions and depths that are statistically different from either the shoulder or 5 cm samples, respectively, based on Tukey-adjusted p values estimated from a generalised linear model (Table S2).”

R3.21. Page 6, Line 13: It doesn't make sense to start this sentence with “however”; isn't this an addition that agrees with the previous result?

These results are contrasting. One shows no change and the second a homogenisation. We modified lines 143-148 as follows to ensure clarity:

“There was no change in the proportion of universal compounds between the shoulder position and streams (Fig. 1d; Table S2), as expected if hydrological mixing was important along the hillslope gradient. However, the DOM pool was similarly homogenised...”

R3.22-23.

Page 6, Line 13-15: I'm lost...generally in this paragraph, I'm finding it hard to corroborate the written results with the figure, and the results are hard to follow. Please improve the reporting of the results.

Page 6, Line 16: Are these percent differences? Also I'm confused where these numbers came from...when I look at the Fig 1e, the range of percentages is less than 5%.

As the Reviewer guessed, we presented % differences and have since added the following text to clarify what we are reporting. This text should align more clearly with the figure.

Added on lines 148-152:

“The relative abundance of universal compounds increased by (95% CI) 9.4% (5.9 to 12.9%, linear model: $t=5.5$, $p < 0.001$, $df=57$) from an estimated mean of 54.3% (51.8 to 56.8%) at 5 cm to 64% (61.4 to 66.0%) at 60 cm (Fig. 1e). There was a similar 8.1% (6.4 to 9.7%, $t=3.0$, $p=0.029$, $df=57$) relative increase in universal compounds from 56.9% (54.4 to 59.3%) at the shoulder to 59.8% (54.9 to 64.5%) at the stream, respectively (Fig. 1f).”

R3.24. Page 6, Lines 17-21: This doesn't follow and seems out of place

What we were trying to convey was that our set-specific “universal” compounds are similar to set-independent universal compounds identified as degradation end-products at the bottom of the ocean:

Added on lines 153-157:

“Universal compounds identified as those occurring in all our samples had similar molecular properties to literature definitions of degradation end-products that were independent of our sample set (Fig. S2). We found similar results when we matched our molecular formulae to those considered universal²² across aquatic ecosystems, again, likely because they reflect end-products of degradation (Table S2).”

R3.27. Page 7, Lines 1-7: These increases are very subtle and do not provide the weight of evidence to claim that this is evidence of progressive re-working of DOM. These are increases of 10 Daltons or a quote “much stronger” increase of 15 Daltons, and are all in the 415-452 Da range. Even if this is statistically significant, it is quite small and I cannot see taking such a strong biological interpretation from this result. The Roth paper that is cited defines low molecular mass as 150-300 Da, mid molecular mass as 300-450Da, and high molecular mass as >450Da. Perhaps there is signal that is lost in taking the average of all the molecules, which would be seen you instead looked at how the abundance of these different size classes changed over the hillslope and depth gradients?

We apologise for the confusion. The Reviewer aptly points out that much of the signal is lost by taking the average of all the molecules (we weight by the intensity of the molecule, i.e. abundance, so ignoring abundance is not the cause of the problem). The increases that we originally reported are consistent with the ranges reported in the Roth paper of 21 Da (6% change), on average. However, rather than classify molecules into arguably arbitrary size bins like Roth et al., we now focus the Main Text on individual compound classes. Here, the effect size was much larger – up to 73 Da (range: 22-73 Da) in size or a 20% (range: 9-20%) change and presented in compound classes that are more biogeochemically relevant.

To address the Reviewer's concern, we have:

1. Removed the paragraph with averaged and the supporting information.
2. Added absolute changes in the intensity-weighted mean molecular mass to the reports of the compound class breakdown lines 190-223.

R3.28. Page 7, Line 12: $n=1216$...what? Compounds?

Thank you for pointing this out. Added “molecular formulae” on line 163.

R3.29. Page 9, Line 13: How many replicates for the means? 4?

Yes, we have added specified 4 replicates on line 170.

R3.30. Page 8, Line 2: So is this the same data, just aggregated differently? If so, this was a bit confusing

We have added the following on lines 168-170 to clarify: “Estimated marginal means \pm 95% CI denoted by black squares were averaged across catchments and either depth or hillslope positions.”

R3.31. Page 8, Line 5: Specify the statistical test

Added on lines 170-173:

“Black stars denote hillslope positions and depths that are statistically different from either the shoulder or 5 cm samples, respectively, based on Tukey-adjusted p values estimated from a generalised linear model (Table S2).”

R3.32-33.

Page 8, Line 7: This statement is very strong and I am not seeing enough evidence to justify it
Page 8, Lines 7-10: I am confused how an *increase* in lignin-like material would indicate an *increase* in microbial reworking. Lignin is a plant biomolecule. I would interpret an increase of lignin-like compounds with depth as a *decrease* in microbial reworking...meaning the plant material is not being degraded and sticking around longer.

In this sentence we have not clearly communicated the idea. We are suggesting lignin-like compounds are increasing in only relative abundance because other compounds are being preferentially degraded, not that microbial activity is making lignin-like compounds. We have modified the wording on lines 175-176 as follows:

“Convergence towards a universal DOM pool was due to the preferential degradation of subsets of compounds actively reworked by microbes.”

R3.34. Page 8, Lines 9-11: This result doesn't appear to be the trend seen in Fig 2, why?

Fig. 2 is displaying the average intensity-weighted mass of non-universals. We have added an additional supplemental Fig. S5 for the universal group which is now referred to on lines 196 and 198.

R.3.35. Page 8, Lines 13-16: This sentence is confusing. I'm guessing the authors are trying to explain the phenomenon of how changes in relative abundance can be caused by other compounds going down, but it's confusing

We agree and have modified the entire paragraph for clarity.

Further, we reworded the topic sentence on lines 175-176, as explained in R3.32-33, to help clarify this concept.

R.3.36. Page 8, Line 20: Please report statistical significance, and provide a figure for these results (either in text or in the supplemental). I initially was very confused because I thought this data was in Fig.2 and the results don't match—because Fig 2 is not showing abundance, it's showing changes in molecular mass. Generally, I think it would be useful to see the same graphs but representing the abundance of the compounds.

We have provided new Figures S5 and S6 and cite them with the corresponding results on lines 196, 198, 213, 216, and 219.

R3.38. Page 9, Line 3: Changed how? Vague, please specify (they increased)

Thank you for pointing this out. We had added “increased” on lines 202.

R3.40. Figure 2: The colors in the legend do not match the image because it appears transparency was applied to the lines (please make them match). “Darkened” lines would be better written as “solid” lines, and the rest are dashed

We have modified the transparency in the legend to match the figure and changed “darkened” to “solid” in the caption.

R3.41-43. Page 11, Lines 2-4: The decrease of C concentrations with depth does not necessarily reflect continuous microbial degradation, it reflects that there are lower amounts of C at depth. This is generally true of most soil depth profiles in mineral soils—soil TOC decreases with depth, and that C is typically older than the surface C. It is more likely that DOC concentrations are correlated with TOC concentrations.

Page 11, Lines 4-5: Also, this is a very normal thing to see in soil profiles—microbial activity and abundance decreases with depth as TOC concentrations decrease. You could cite some papers and do an “as expected” part here

Page 11, Lines 16-17: I am seeing no evidence here of a degradation cascade, let alone two degradation cascades. I see some very normal patterns for soil depth profiles taken from mineral soils. And no metabolic pathways are discussed. Please revise and provide a more tempered conclusion that reflects the data presented.

We have revised this paragraph in line with the Reviewer’s three comments above. We cited papers to emphasise that these are “expected” trends, thereby addressing the first two comments. To address the third comment, we clarified the conclusion: there are clear environmental differences between the depth and hillslope gradients, yet, a similar convergence in molecular composition of DOM, as explained in previous paragraphs. We suggest that microbial processing may contribute to this convergence (the convergence is evidence of the degradation cascade), and expand on this idea in the next paragraph.

R3.44. Page 11, Lines 6-7: Why isn’t the size-exclusion chromatography data presented in the main paper, and how are you defining plant- vs microbial-derived C?

We mistakenly omitted the reference pointing to the LC-OCD data in the Main Text. The data are presented in Figure 3c, which we have now cited. We have also included the HMWS data in the Supplement (Table S1) to address R3.14. We removed the reference to plant- vs microbial-derived C from the text.

R3.45. Page 11, Lines 12-15: I would reverse the order of these two pieces of information; present the result first (decline in proteins), followed by interpretation.

Thank you for the helpful suggestion. We have reversed their order. On lines 245-252: “Alongside the evidence of microbial consumption of dissolved organic carbon (Fig. 3a), nitrogen-rich proteins identified by FT-ICR-MS strongly declined from shoulder to toeslope (Fig. 2b). Nitrogen in the form of proteins was likely selectively adsorbed by clays⁴⁴ that accumulate at the bottom of hillslopes⁴⁵.”

R3.46. Figure: 3: How do you have C:N values close to zero?

We made an error in not specifying that this is organic carbon. We have added this specification. We also note that the ratio of C:N (in various C or N forms) is very low at 60cm depth where we effectively measured the leachate of wet N-saturated clays. N-saturated clays are known to have low C:N ratios less than 5 with examples of river bed sediments with a TOC: TN ratio of 2.6 e.g. N-saturated clays:

<https://www.sciencedirect.com/science/article/pii/S0016706122003330>, riverbed:

<https://agupubs.onlinelibrary.wiley.com/doi/full/10.1029/2018JG004674>. We have also modified the y-axis so that it begins at 1 rather than 0 so it is clear that all the lowest values are greater than one (and close to 2).

R3.47. Why not show these as regressions and provide r-squared, etc?

These are indeed the results of generalised mixed models, i.e. regressions. All the regression coefficients and R^2 values are given in Table S7, which we now cite on lines 239, 243, and 266.

R3.48. And finally, what is the statistical test, and do your p values account for multiple comparisons? If not, a Tukey post-hoc test would be more appropriate.

We have added this specification on lines 263-266:

“Black stars denote hillslope positions and depths that are statistically different from either the shoulder or 5 cm samples, respectively, based on Tukey-adjusted p values estimated from a generalised linear model (Table S7).”

R3.49-50

Page 13, Line 4: What are these environmental variables? Are these different molecule categories, in addition to edaphic factors like pH etc? This is a large number of variables

Page 13, Line 19: This is a lot of variables! Are these variables edaphic factors or molecular classes?

We have clarified the types of variables in the Main Text as suggested by the Reviewer and included a new Table S8 that lists all the variables and distinguishes those that are important for explaining DOM composition across the 8 compound classes, i.e. the 62 variables listed on the original line 19.

Added on lines 273-275:

“We measured 475 environmental variables related to the gene activity of carbohydrate-degrading enzymes, the activity of extracellular enzymes or level of carbohydrate substrate utilisation, and soil physico-chemical conditions (Table S8).”

Added on lines 289-290:

“This analysis identified 62 environmental variables that were important for explaining DOM composition across the 8 compound classes (Table S8).”

Page 13, Line 11: I don't understand what you mean by “it was comparable on a per-variable-basis” as a justification for them still being important even though they're only explaining <4% of the variation...

The argument is that the spatial variables are at least as important as the biological variables if one adjusts for the number of variables in each category. However, we have decided instead to remove this text.

R3.52. Page 13, Line 13: What variables explained the most variation?

This information is included in Fig. 4b as so we have referenced it on line 294 to indicate this.

R3.53. Page 14, entire paragraph: I would like the authors to provide citations for the activities of these enzyme classes. Also, many of these enzyme classes contain enzymes that can act on many types of substrates and can have plant/fungal/or bacterial origin. I would like more support for their assertions.

We have added citations 46-49 for the activities of the enzyme classes and these have been screened to ensure that the activities of these enzymes are microbial in origin.

3.54. Page 14, Line 1: these CAZymes can also potentially break down microbial carbohydrates

Yes, CAZymes can also potentially break down microbial carbohydrates. However, the CAZymes discussed in this paragraph are primarily known to break down plant-derived carbohydrates, and we have added references to support these claims.

R3.55. Page 14, Line 5: I'd change "these genes" to "these CAZymes" to make it clear you're specifying the CAZymes from the previous sentence. Or you could use a semicolon, but then the sentence is very long

Thank you for the helpful suggestion. We have changed "these genes" to "these CAZymes".

R3.56. Page 14, Line 10: What do you mean by "at a community-level"? Are you talking about bulk measurements from enzyme activity assays?

Yes, we are referring to the bulk measurements from enzyme activity assays and have clarified on lines 303-305:

"We also found evidence that these enzymes were expressed at a community-level as measured from enzyme activity assays."

R3.57. Figure 4: Fig 4b "activity" misspelled. I'm a bit confused about the different sets of totals—please clarify your presentation. Do all the totals add up to 100%? For instance, for the Unsat Hydrocarbons, there is a total variation explained by model (0.66...66%?), and Envi Variation (maybe 95%?), 4% spatial variation, and then Gene Activity (66%), Enzyme activity (25%) and Physical Chemistry (8%). I'm guessing that everything in Fig 4b represents the Envi Variation? (so it's actually like...of the 66% of total variation explained by the model, Gene activity represents 43% of the variation (66% of the total 66% of variation explained by the model).

Fixed the typo. We have modified the figure, changing the label y-axis in panel (a) to indicate that this is the relative variation explained (out of 100%). We have also modified the figure caption to clarify the presentation:

"... a. The relative variation explained by spatial and environmental variables in a variance partition analysis of the relative abundances of non-universal-molecular formula in each compound class. Above each bar is the absolute variation (%) explained by the model."

Page 16, Lines 9-10: Alright, I'm not seeing this. What data showed that you switched from plant processing genes to microbial processing genes? Also you don't show any gene expression across the gradients; you're showing what % of the variation in the model for a particular class of compounds is explained by different CAZymes.

We agree with the Reviewer that, until this point, we have not shown any gene expression across the gradients. However, the lines that the Reviewer highlights are the topic sentence of a new paragraph, which we go on to support in subsequent sentences. The subsequent sentences describe analyses of gene expression across the gradient, summarised with statistical results presented and cited in Tables S9 and S10. These analyses of gene expression show that a small number of genes increase (those associated with processing microbial-derived OM) and a small number decrease (those associated with degrading plant-derived OM) across the gradients. For this reason, we believe this sentence is supported and have not made further changes to the text, especially as the Reviewer seems to agree with this interpretation in their comment below R3.59.

R3.58. Page 16, Line 9: What is the justification for using "universally" here?

We have removed the word "universally".

R3.59. Page 17, Line 5: I agree that the data suggests that microbes are processing compounds of microbial origin rather than plant origin at depth. However, the phrase of “transforming plant-derived compounds into higher molecular weight [compounds]” makes it sound like there are plant inputs down there, and that is not what your data suggests. That messaging is very confusing. At depth, the source is microbial. I suggest you rephrase.

Thank you for pointing this out. We have modified lines 347-349 to read:

“These results further implicate microbes in transforming plant-derived compounds from shallower depths or higher hillslope positions into larger molecular weight compounds at later positions along degradation cascades.”

R3.60. Page 17, Line 10: What is the evidence for removing the source-specific components? Have clarified on lines 352-353 as follows: “Our study provides new evidence that DOM converges towards a universal compound pool as microbial reworking removes components typical of shallower soil depth and higher hillslope positions.”

R3.61. Fig 5: I don't understand the y-axis. Is this percent similarity to deep sea samples? If so, change the unit from (%) to (% similarity).

Modified. Thank you.

R3.62. Page 17, Line 20: I would flesh this out a little bit more (e.g., deep soils and toe/stream were most similar to deep sea). Though the error bars are pretty large...and surface soil is still very similar, so it increases but only from 39% to maybe 43%, not accounting for error bars. The Hillslope position looks clearer. As I asked before, is this the same data presented twice, but just aggregated differently? (by either hillslope or depth) That should be explained

Changed on lines 362-365:

“We subsequently found samples expected to be exposed to microbial processing for longer, that is, further along the land-ocean continuum, such as deep soils and toeslope and stream positions, were most like deep-sea DOM (Fig. 5).”

We have addressed their other point as in R3.30 and have clarified the figure caption by adding on lines 385-386: “Results for soil depth and hillslope position were averaged across sites for each respective gradient.”

R3.63. Page 19, Line 5: How does persistence follow here?

Changed on lines 388-291:

“Because compounds vary in their persistence, by demonstrating that microbial processing changes the abundance of individual compounds along different environmental gradients, our work advances our understanding of how organic compounds accumulate and influence soil carbon sequestration.”

R3.64. Page 19, Line 8: Which were the non-universal? Reminder would be helpful

Changed on lines 393-395:

“Second, non-universal compounds (for example, in the carbohydrate-like class) were consistently reworked by microbes.”

R3.65. Page 19, Line 13: Previously you were saying (and I think correctly), that pectate lyases are associated with plant material and decreased over the soil profiles. It doesn't make sense for it to be associated with general degradation processes

By “general”, we were referring to degrading plant material, albeit in a general sense, rather than all organic matter. We have altered the text to be more precise on lines 397-400:

“Given the ubiquity of plant cell walls in nature, these results suggest widespread microbial metabolic pathways, like those involved in the production of pectate lyases²⁴, could underpin a general land-to-ocean degradation process.”

R3.66. Page 19, Lines 14-15: You did not present data on the taxonomic composition of pectate lyases, but you appear to be invoking functional redundancy here. You cannot say that pectate lyase mediated degradation is independent of microbial taxonomic composition unless you test this.

We agree and have tempered this sentence to suggest it “could persist” as a possible mechanism independent of taxonomic composition.

R3.67. Page 19, Lines 17-18: What consistent patterns are you citing? Be more specific. It sounds like you're talking about CAZyme data; if so, what consistent patterns within that? Changed on lines 402-405: “Even if environmental conditions were modulated along the land-ocean continuum by time or other unmeasured environmental factors, the consistent **convergence toward a universal compound pool** that we observed across different environments imply a general degradation process.”

R3.68. Page 19, Line 22: Change “individuals” to “individual”

Thank you for the sharp eye. Corrected.

R3.69. Page 19, Line 23: Vague, what about your methods would allow you to test this?

We replaced “our methods” with what our specific methods on lines 410-411:

“...and our paired FT-ICR-MS, shotgun metatranscriptomic sequencing and metabolic methods approach...”

R3.70. Page 19, Line 24: What have you presented that would help you estimate the degradation state?

We have reworded on lines 411-413:

“More generally, the reasons for the spatial variation in the degradation state of DOM that we identified here can improve soil-to-stream carbon management.”

Reviewer #2 (Remarks to the Author):

This reworked manuscript address my concerns of the first draft.

Reviewer #3 (Remarks to the Author):

The authors have addressed many of my concerns, and I appreciate the detail they provided in their letter regarding those changes.

Generally, I would find it helpful if the authors clarified what they mean by "universal pool" in more places – I've suggested some locations and possibly phrasing in my "additional comments" below.

However, a major point of confusion for me was that in many places, it reads like the microbes are *producing* the universal compounds, when it seems like there are actually a few things going on:

1. Microbes are consuming non-universal compounds, which both transforms them into more non-universal compounds, and also depletes non-universal compounds from DOM.
2. Universal compounds resemble plant compounds (e.g., lignin-like, tannin-like) that are likely difficult to decompose. These could have accumulated because microbes never started decomposition, microbial degradation was slower than the production and transport of these compounds, OR microbial depolymerization was incomplete and decomposition halted for some reason. Since these appear to be derived from plants, they are not microbially produced; at best, they would be partly depolymerized or decomposed, i.e., microbially modified.

For instance, these lines make it sound like the microbes are producing the universal compounds de novo:

- In the abstract (lines 36-37): "Our results implicate continuous microbial reworking in shifting DOM towards universal compounds in soils"
- At the last line of the Introduction (lines 115-116): "Our results now implicate that common metabolic processes shift DOM towards homogenized compounds along a soil headwater continuum..."

I would like a more nuanced version of these statements in the paper, which I think reflects an important clarification on the mechanisms generating the universal pool. This does not simply appear to be an enzymatic degradation cascade where microbial products are reworked and reworked until they somehow become unusable—rather it appears universal and non-universal pools are undergoing two different sets of processes, where the universal pool retains plant-like characteristics (possibly because decomposition didn't commence or was terminated). The authors described their hypothesized mechanisms at lines 391-395, which make sense to me: 1. Selectively retained due to energetic costs for degradation, 2. Microbes rework non-universal compounds (thus depleting particular compounds from DOM). A possible rephrasing of the second sentence could be that would reflect both these mechanisms could be: "Our results now implicate that microbes transform non-universal compounds, leaving behind a universal pool of compounds throughout the soil headwater continuum..."

I do still have questions about the transcriptome (see my response to R3.27 below), but the enzyme activities and transcriptome appears to primarily shed light on the *non-universal* pool (carbohydrate, unsaturated hydrocarbon). Plant material is clearly degraded and generates a large pool of non-universal products. The universal pool is not discussed for the transcriptome, which seems odd (maybe I've missed it?). My guess would be that universal products are produced at a very early part of plant-litter decomposition so they retain their plant-like characteristics (i.e., not at the end of a cascade), and easily degradable components of that plant material go on to be highly cycled and become part of the non-universal pool—but I would like to hear the authors' opinion on this.

Response to a few points from the rebuttal:

R3.21 The authors assert that these two sentences are contrasting, but to my read, this still does not seem to be the case. In the previous sentence, hydrological mixing (a homogenization) caused similar proportions (or “no change” in the proportions) of the universal compounds across sites. The next sentence says the DOM pool was similarly homogenized—also due to hydrological mixing? If my interpretation agrees with yours, I would simply delete the word “however” to indicate that the homogenized DOM supports or echoes the previous result (similar DOM across sites). If there is another mechanism causing DOM homogenization, I would call that out, so it is clear what is contrasting. Or maybe it’s the word “homogenization” that’s the issue?

R3.22-23 I appreciate the thorough addition of stats throughout the document. However, the placement of the (95% CI) before the percentage is slightly confusing; it would be better following the percentage and immediately before the range, e.g., “9.4% (95% CI: 5.9%-12.9%, linear model...”

R3.27 My point was not understood by the authors, so it was not addressed. Let me clarify. I wanted to see the gene expression data presented at a finer scale; I cannot validate the authors’ claims without seeing this. For example, in the Supplemental Tables, the only value presented is the mean value for the *entire gradient* (averaging across all ALL depths or hillslope positions). Instead, I wanted to see mean expression for each gene of interest (e.g., PL1, AA1, GH15, GH51, GH135) at *each* depth and hillslope position (e.g., using a heatmap or table, show mean PL1/AA1/etc expression across the depth gradient at 5cm, 15cm, 30cm, 60cm). I tried to calculate the mean increases or decreases in gene expression that were presented in the text and could not do this; I was unclear about what these numbers meant and what magnitude we’re talking about.

Please provide the following information:

- It looks like you used a model that asked whether depth or hillslope was important variable structuring the data (e.g., CAZy_expression ~ depth), but that does not mean there were consistent increases or decreases across a depth. One depth could have had very high gene expression relative to the other depths.
- How did you handle random effects? (depths for a single core are not independent measurements; they would be dependent on the depths above and below). Did you “nest” the data by core?
- Lines 338-341: Did AA1 increase or decrease across the gradient? The text does not specify

R3.59 At L349, I would go one step further to clarify and change the phrase to: “larger molecular weight compounds of microbial origin”

R3.66 This still reads as an unfounded speculation (Line 400-401) since you don’t provide evidence to support it, and don’t provide rationale for why it’s an important point. Also, the concept of mineral protection is introduced out of the blue (why were these two ideas paired together, anyways?). I would simply delete this sentence since it doesn’t advance your argument. If you feel it is an important point to make, I would explicitly state “We hypothesize that...” and provide some previous literature or rationale to back up why this is an important point.

R3.67 The clarification does help, but “imply a general degradation process” seems like a stretch. Decomposition is accomplished by a large number of enzymes, which you show; biologically, the degradation process is not one enzyme or one microbe, or even one class of reactions. Also, pectate lyases explained the variation in non-universal compounds, rather than universal compounds. Please clarify.

Additional comments:

Line 33 Add “non-universal” before carbohydrate

Line 65 This sentence is unfinished

Lines 102-115 I found this confusing, would be good to tighten up. There are supposedly two predictions, but there are many more predictions here

Line 105 Change period to semicolon?

Line 140 For parenthetical citation, specify which sub-figure you're talking about (a, b, c...)

Line 162-163 In the first Figure caption, it would be useful to define "universal" to the reader. For example, you could add to the end of the sentence "...which is defined as compounds that are present in all samples."

Line 176 I would clarify by adding a statement like this to the end of the sentence: "...microbes, leaving behind a ubiquitous pool of compounds observed across all samples"

Line 228 By "removed," do you mean de-emphasized by graying them out? Or are the lignins and tannins in this figure "non-universal" lignins and tannins? I'm guessing the latter

Line 287 What do you mean here by degradation cascades? Are you talking about depth/hillslope or enzymatic cascades? In this context, the headwater-ocean continuum makes more sense to me

Line 297 Add non-universal: "For non-universal carbohydrate..."

Line 325 Add a parenthetical reference to Fig. 2 at the end of this sentence to cite your data

Line 328 Weak reference: change "its" to "the microbiome's"

Line 332 Add "non-universal" before classes

Line 333 To help the reader, I'd change the parenthetical citation to "(Fig. 4b, bolded compound classes)." This would have saved me some effort and a trip back to Fig 2

Line 354 I would clarify and add to the ending of this sentence: "...positions, and leaves behind a ubiquitous pool of lignin- and tannin-like compounds found across all samples (Fig. 1b)."

Line 397 I would add a parenthetical citation at the end of this sentence to Fig 2

Line 399-400 A general process...for non-universal compounds?

Reviewer #2 (Remarks to the Author):

This reworked manuscript address my concerns of the first draft.

We thank the Reviewer and no further changes are required based on their comments.

Reviewer #3 (Remarks to the Author):

The authors have addressed many of my concerns, and I appreciate the detail they provided in their letter regarding those changes. Thank you for appreciating our work.

Generally, I would find it helpful if the authors clarified what they mean by “universal pool” in more places – I’ve suggested some locations and possibly phrasing in my “additional comments” below.

We have clarified what we mean by “universal pool” using the phrasing suggested by the Reviewer in all the locations that the Reviewer identifies. We provide the specific changes to the text / line numbers below in response to the Reviewer’s *Additional comments*.

However, a major point of confusion for me was that in many places, it reads like the microbes are *producing* the universal compounds, when it seems like there are actually a few things going on:

1. Microbes are consuming non-universal compounds, which both transforms them into more non-universal compounds, and also depletes non-universal compounds from DOM.
2. Universal compounds resemble plant compounds (e.g., lignin-like, tannin-like) that are likely difficult to decompose. These could have accumulated because microbes never started decomposition, microbial degradation was slower than the production and transport of these compounds, OR microbial depolymerization was incomplete and decomposition halted for some reason. Since these appear to be derived from plants, they are not microbially produced; at best, they would be partly depolymerized or decomposed, i.e., microbially modified.

For instance, these lines make it sound like the microbes are producing the universal compounds de novo:

- In the abstract (lines 36-37): “Our results implicate continuous microbial reworking in shifting DOM towards universal compounds in soils”
- At the last line of the Introduction (lines 115-116): “Our results now implicate that common metabolic processes shift DOM towards homogenized compounds along a soil headwater continuum...”

I would like a more nuanced version of these statements in the paper, which I think reflects an important clarification on the mechanisms generating the universal pool. This does not simply appear to be an enzymatic degradation cascade where microbial products are reworked and reworked until they somehow become unusable—rather it appears universal and non-universal pools are undergoing two different sets of processes, where the universal pool retains plant-like characteristics (possibly because decomposition didn’t commence or was terminated). The authors described their hypothesized mechanisms at lines 391-395, which make sense to me: 1. Selectively retained due to energetic costs for degradation, 2. Microbes rework non-universal compounds (thus depleting particular compounds from DOM). A possible rephrasing of the second sentence could be that would reflect both these

mechanisms could be: “Our results now implicate that microbes transform non-universal compounds, leaving behind a universal pool of compounds throughout the soil headwater continuum...”

We thank the Reviewer for their elegant and nuanced phrasing of the processes. We have tried to incorporate their phrasing on the lines they identify above (original lines 36-7, 115-116). This issue does not crop up anywhere else in the Main Text.

Changed on lines 36-7:

“Our results suggest that microbes transform non-universal compounds, leaving behind a universal pool of DOM.”

Changed on lines 115-18:

“Our results now implicate common microbial metabolic processes in transforming non-universal compounds, leaving behind a universal pool of DOM along the soil-headwater continuum...”

I do still have questions about the transcriptome (see my response to R3.27 below), but the enzyme activities and transcriptome appears to primarily shed light on the *non-universal* pool (carbohydrate, unsaturated hydrocarbon). Plant material is clearly degraded and generates a large pool of non-universal products. The universal pool is not discussed for the transcriptome, which seems odd (maybe I've missed it?). My guess would be that universal products are produced at a very early part of plant-litter decomposition so they retain their plant-like characteristics (i.e., not at the end of a cascade), and easily degradable components of that plant material go on to be highly cycled and become part of the non-universal pool—but I would like to hear the authors' opinion on this.

We have not discussed the transcriptome in the context of the universal pool because this would not produce additional information. FT-ICR-MS is a semi-quantitative technique, and to standardise values across samples we scale signal intensities of each compound relative to the total intensity of the sample. Therefore, an increase in the proportion of universal compounds or universal compound signal intensity corresponds to a decrease in non-universal compounds and vice-versa. *We have added the following text on lines 297-99:* “As the relative intensity of non-universal compounds is the inverse of universal compounds, we did not repeat the variance partition analysis with the universal pool.”

We chose to focus on the non-universal pool because there were no universal carbohydrates (as shown in Fig 1b), so an analysis of carbohydrate-active enzymes in the context of the universal pool did not make sense.

Response to a few points from the rebuttal:

R3.21 The authors assert that these two sentences are contrasting, but to my read, this still does not seem to be the case. In the previous sentence, hydrological mixing (a homogenization) caused similar proportions (or “no change” in the proportions) of the universal compounds across sites. The next sentence says the DOM pool was similarly homogenized—also due to hydrological mixing? If my interpretation agrees with yours, I would simply delete the word “however” to indicate that the homogenized DOM supports or echoes the previous result (similar DOM across sites). If there is another mechanism causing

DOM homogenization, I would call that out, so it is clear what is contrasting. Or maybe it's the word "homogenization" that's the issue?

The contrast is between the measured variables: the proportion of universal compounds versus their relative abundance. In the first case, there is a difference between depth and position gradients, and in the latter there is not. To clarify we have reordered the clauses in the sentence:

"However, when we measured the proportion of signal intensity (i.e., relative abundance) attributed to universal compounds, the DOM pool was similarly homogenised along both soil depth and hillslope gradients."

R3.22-23 I appreciate the thorough addition of stats throughout the document. However, the placement of the (95% CI) before the percentage is slightly confusing; it would be better following the percentage and immediately before the range, e.g., "9.4% (95% CI: 5.9%-12.9%, linear model..."

Thank you for this suggestion. We have changed all instances exactly as suggested.

R3.27 My point was not understood by the authors, so it was not addressed. Let me clarify. I wanted to see the gene expression data presented at a finer scale; I cannot validate the authors' claims without seeing this. For example, in the Supplemental Tables, the only value presented is the mean value for the *entire gradient* (averaging across all ALL depths or hillslope positions). Instead, I wanted to see mean expression for each gene of interest (e.g., PL1, AA1, GH15, GH51, GH135) at *each* depth and hillslope position (e.g., using a heatmap or table, show mean PL1/AA1/etc expression across the depth gradient at 5cm, 15cm, 30cm, 60cm). I tried to calculate the mean increases or decreases in gene expression that were presented in the text and could not do this; I was unclear about what these numbers meant and what magnitude we're talking about.

We have added a heatmap as Fig. S5 with the genes of interest, that is, the genes that vary significantly along either depth or hillslope gradients. The heatmap has been presented in the format described by the Reviewer.

Please provide the following information:

- It looks like you used a model that asked whether depth or hillslope was important variable structuring the data (e.g., CAZy_expression ~ depth), but that does not mean there were consistent increases or decreases across a depth. One depth could have had very high gene expression relative to the other depths.

Expression at each depth and hillslope position is now provided in the heatmap, allowing the reader to determine if there were consistent increases or decreases across a given gradient.

However, our hypotheses and conclusions do not rely on, or even mention the need for, consistent changes in gene expression. For example, some genes may increase in expression from 5 to 15 cm, and a complementary gene, such as one that is part of the same wider biochemical pathway, may take over decomposition and increase in expression from 15 to 60 cm. Our aim with this analysis was simply to explore which of hundreds of genes could be tied to changes in organic matter decomposition. We have further addressed this possibility by including an ordination analysis in the Main Text.

Added on lines 351-61:

“Because genes also interact, expression patterns can shift across the spatial gradients without necessarily showing monotonic increases. Therefore, we also used principal component analysis (PCA) to visualise differences in gene expression along the spatial gradients. Both depth and hillslope explained variation in gene expression as they were correlated with the first two axes of the PCA ($r^2=0.11$, $p=0.029$ and $r^2=0.15$, $p=0.047$, respectively). At later stages of the degradation cascades, expression shifted towards CAZymes that degrade microbial-derived OM, e.g., GH23, GH135, and CBM2 (Fig. S6). Together, these results further implicate microbes in transforming plant-derived compounds from shallower depths or higher hillslope positions into larger molecular weight compounds microbial origin at later positions along degradation cascades.”

•How did you handle random effects? (depths for a single core are not independent measurements; they would be dependent on the depths above and below). Did you “nest” the data by core?

We thank the Reviewer for raising this important point. The Reviewer is correct that depths are not independent. For this reason, we have carefully reflected on this analysis and extensively revised the underlying approach to address their comment. While there are some changes in the specific CAZymes that are identified, our main conclusions are unchanged.

Added on lines 647-51:

“To test differential expression of genes annotated as CAZymes, we fitted mixed effects models to raw transcript counts with negative binomial error structures using the R package *glmmSeq*⁹⁰. Models were fitted to each CAZyme individually with soil depth, hillslope position, and catchment identity as predictors. To account for variation from repeated soil core measurements, we added core identity as a random effect.”

• Lines 338-341: Did AA1 increase or decrease across the gradient? The text does not specify This sentence has been removed so this comment is no longer relevant. For other CAZymes that changed across the gradients, we now state explicitly if they increased or decreased.

R3.59 At L349, I would go one step further to clarify and change the phrase to: “larger molecular weight compounds of microbial origin” **Changed exactly as suggested.**

R3.66 This is still reads as an unfounded speculation (Line 400-401) since you don't provide evidence to support it, and don't provide rationale for why it's an important point. Also, the concept of mineral protection is introduced out of the blue (why were these two ideas paired together, anyways?). I would simply delete this sentence since it doesn't advance your argument. If you feel it is an important point to make, I would explicitly state “We hypothesize that...” and provide some previous literature or rationale to back up why this is an important point. **We have deleted this sentence as suggested by the Reviewer.**

R3.67 The clarification does help, but “imply a general degradation process” seems like a stretch. Decomposition is accomplished by a large number of enzymes, which you show; biologically, the degradation process is not one enzyme or one microbe, or even one class of reactions. Also, pectate lyases explained the variation in non-universal compounds, rather

than universal compounds. Please clarify.

We have removed this sentence so this comment is no longer relevant.

Additional comments:

Line 33 Add “non-universal” before carbohydrate **Added.**

Line 65 This sentence is unfinished

Thanks for catching this. We should have inserted a comma rather than period.

Lines 102-115 I found this confusing, would be good to tighten up. There are supposedly two predictions, but there are many more predictions here

Thanks for pointing out this discrepancy. We rephrased the text to state that we have “two contrasting hypotheses”, and then explained the “predictions” associated with each “hypothesis”. For the first hypothesis, we have two separate predictions between the depth and hillslope gradient, whereas we have only one prediction from the second hypothesis.

Line 105 Change period to semicolon?

We agree that the connection between the two sentences needs to be improved. As we prefer to keep sentences concise, we have added “therefore” on line 105 so that the predictions logically follow each other. This structure mirrors that used for the next prediction of this hypothesis.

Line 140 For parenthetical citation, specify which sub-figure you're talking about (a, b, c...)

We have removed this citation to the figure.

Line 162-163 In the first Figure caption, it would be useful to define “universal” to the reader. For example, you could add to the end of the sentence “...which is defined as compounds that are present in all samples.”

Added on lines 165-66:

“c. to f. Composition of universal compounds, which are defined as those present in all samples.”

Line 176 I would clarify by adding a statement like this to the end of the sentence:

“...microbes, leaving behind a ubiquitous pool of compounds observed across all samples”

Changed on lines 177-78:

“Convergence towards a universal DOM pool was due to the preferential degradation of subsets of compounds actively reworked by microbes, leaving behind the universal pool of compounds that was observed across all samples.”

Line 228 By “removed,” do you mean de-emphasized by graying them out? Or are the lignins and tannins in this figure “non-universal” lignins and tannins? I'm guessing the latter

Changed on lines 228-30:

“Mean estimated molecular mass (\pm 95% CI) of DOM for each compound class when universal compounds were removed from calculations, that is, we plot only non-universal compounds.”

Line 287 What do you mean here by degradation cascades? Are you talking about depth/hillslope or enzymatic cascades? In this context, the headwater-ocean continuum makes more sense to me

We have replaced “degradation cascades” with “soil depth and hillslope gradients”.

Line 297 Add non-universal: “For non-universal carbohydrate...” **Added**

Line 325 Add a parenthetical reference to Fig. 2 at the end of this sentence to cite your data **Added**

Line 328 Weak reference: change “its” to “the microbiome’s” **Changed**

Line 332 Add “non-universal” before classes **Added**

Line 333 To help the reader, I’d change the parenthetical citation to “(Fig. 4b, bolded compound classes).” This would have saved me some effort and a trip back to Fig 2

Added

Line 354 I would clarify and add to the ending of this sentence: “...positions, and leaves behind a ubiquitous pool of lignin- and tannin-like compounds found across all samples (Fig. 1b).”

We appreciate the Reviewer’s suggestion but wanted to avoid a sentence that would span four lines of text. We have therefore revised the sentence as closely as possible to the Reviewer’s suggestion while trying to remain concise.

Changed lines 363-65:

“Our study provides new evidence that DOM converges towards a universal compound pool dominated by lignin- and tannin-like compounds that are left behind as microbial reworking removes components typical of shallower soil depth and higher hillslope positions.”

Line 397 I would add a parenthetical citation at the end of this sentence to Fig 2

We think it is more appropriate to cite Fig 4 here as the subject of this sentence is “genes encoding enzymes involved in breakdown of plant cell walls”, shown in Fig. 4, and their correspondence to compound classes that shift in mass, which is indicated in Fig. 4 by the bolding of compound classes. Thus, all the information to support this statement is captured in Fig. 4.

Line 399-400 A general process...for non-universal compounds?

We hope it is OK not to modify this sentence. We think adding “non-universal” would complicate the sentence and the message that we would like to communicate. The point that we would like to make is that the degradation process – irrespective of whether it arises through changes in universal or non-universal compounds – occurs because of changes in common microbial metabolic pathways. In other words, the focus of this sentence is on common metabolic pathways and their link to degradation rather than the specific types of compounds that are changing.